# ASAP: Adaptive Sliding Agnostic Poisoning Attack on Federated Learning

## Abstract

The primary risk in the federated learning (FL) framework arises from the potential for manipulating local training data and updates, known as a poisoning attack. Among various attack strategies, agnostic attacks have emerged as a significant category that attempts to operate without explicit knowledge of the server's aggregation rules (AGRs). However, existing AGR-agnostic attacks still suffer from a critical dependency: they rely heavily on staying inside the natural per-coordinate variance of honest client updates. These attacks typically operate by analyzing benign clients' gradient patterns, statistical properties, and behavioral characteristics to strategically position their malicious updates. Therefore, to overcome these fundamental limitations of current AGR-agnostic attacks, this work presents the Adaptive Sliding Agnostic Poisoning Attack (ASAP) on FL, which can adaptively, robustly and precisely manipulate the degree of poisoning without the knowledge of AGRs algorithm of the server. Instead of relying on benign client patterns, ASAP incorporates Adaptive Sliding Model Control (ASMC) theory — a sophisticated robust nonlinear control framework that enables adaptive attack. We implement our attack through comprehensive experiments on state-of-the-art (SOTA) Byzantine-robust federated learning methods using real-world datasets. These evaluations reveal that ASAP significantly outperforms all existing agnostic attacks while maintaining complete independence from benign client information, representing a fundamental advancement in FL attack strategies.

## 1 Introduction

The distributed diagram of Federated Learning (FL) ensures training models among clients devices without sharing local data but only sending model updates to a central server (Li et al., 2021). The central server initially sends the global model to selected clients, and each client then trains the model locally using its own data. The locally updated models are then transmitted to the central server, which applies a specified aggregation rule (AGR) to compute the next global model.

However, distributed systems are susceptible to poisoning attacks including both data poisoning attacks (Tolpegin et al., 2020) and model poisoning attacks (Panda et al., 2022) due to its natural mechanism. Most poisoning attacks are designed relying on knowledge of the server's aggregation rules, which is typically difficult to obtain in practical scenarios. Therefore, the development of AGR-agnostic attacks enables attack deployment without aggregation rule awareness or specification. Current AGR-agnostic attack methods, such as LIE (Baruch et al., 2019), depend on estimating statistical properties of benign client updates, particularly coordinate-wise mean $\mu$ and standard deviation $\sigma$, to generate small noises in malicious gradient updates to prevent the optimal convergence. Furthermore, Min-Max and Min-Sum (Shejwalkar & Houmansadr, 2021) attacks constrain the malicious update to lie inside the benign cluster, using a max-distance or sum-of-distances bound, while pushing in an adversarial direction. The fundamental constraint of those AGR-agnostic attacks is their need to remain indistinguishable from benign updates. Moreover, these methods aim to maximize deviation of the global model from optimal convergence, resulting in dynamics that converge to biased equilibrium. Consequently, the estimation of benign clients updates or their statistical properties remains a mandatory requirement for local malicious devices, even though aggregation rule knowledge is no longer required.

To overcome these limitations, we propose Adaptive Sliding Poisoning Attack (ASAP) on FL, a novel FL attack framework that operates without prior knowledge of server aggregation rules or benign client information. The method leverages a combination of Adaptive Sliding Mode Control (ASMC) theory and Fourier series approximation (Young et al., 1999; Ge et al., 1997; Huang & Kuo, 2001). Moreover, ASAP provides precise attack control through adjustable convergence rates and flexible attack objectives capabilities.

To achieve both AGR-agnosticism and precise attack control, we consider the entire FL process as a dynamical system and introduce an adaptive law to estimate the unknown information from malicious clients, along with a control law that guides the global model towards a specified poisoned reference. In particular, we employ Adaptive Sliding Mode Control (ASMC)—an adaptive robust control framework designed for nonlinear systems with uncertain dynamics—which exhibits strong resilience to parameter variations and external disturbances. This eliminates the reliance on explicit knowledge of the AGRs or benign client updates. By employing a Fourier series approximation, the unknown AGR behaviors and benign update patterns are treated as system uncertainties and approximated using a finite number of orthonormal basis functions. ASMC then ensures that the system state converges to and remains on a predefined sliding manifold, thereby enforcing the alignment of the global model with the attack objective. Rather than mimicking benign updates or exploiting AGR structures, the proposed ASAP attack observes the uncertainty from local malicious clients and directly manipulates the malicious gradients to achieve the desired poisoning effect without any prior access to AGR algorithms or benign statistics. Furthermore, ASAP provides fine-grained control over the attack convergence rate, enabling persistent, adaptive, and target-driven manipulation throughout the poisoning process.

Our key contribution can be concluded as below:

- We introduce ASAP, a novel adaptive AGR-agnostic controllable attack that dynamically achieves attack objectives without requiring prior knowledge of aggregation mechanisms.

- ASAP operates without knowledge of server aggregation rules or benign client statistics by leveraging Adaptive Sliding Mode Control (ASMC), distinguishing it from existing agnostic attacks that rely on coordinate-wise estimations and distance constraints.

- We provide theoretical analysis proving that ASAP achieves precise control of attack objectives and converges to predefined targets within finite time at controlled speeds, regardless of the underlying AGR algorithm. Extensive experiments on benchmark datasets against multiple robust AGRs demonstrate consistent superiority over current state-of-the-art (SOTA) methods.

## 2 RELATED WORK

### 2.1 FEDERATED LEARNING

McMahan et al. (2017) firstly demonstrated the algorithm of federated learning (FL) and introduced the FL paradigm. In a typical FL system, a central server coordinates $N$ clients, where client $i$ ($i \in [1, N]$) holds its local private dataset $D_i$ drawn from an underlying distribution $D$, and the datasets can be independently and identically distributed (IID) or statistically heterogeneous (Non-IID). Let $g_t \in \mathbb{R}^r$ denote the global model at iteration $t$, and $g_{\{t,i\}} \in \mathbb{R}^r$ denote the corresponding local model on each client $i$. The objective of FL is to address the optimization problem:

$$g^* = \arg\min_{g_{\{t,i\}}} \frac{1}{N} \sum_{i \in [1,N]} L(g_{\{t,i\}}, D_i), \tag{1}$$

where $L$ denotes the loss function, $g^* \in \mathbb{R}^r$ is the optimal global model, and $r$ represents dimensionality of the parameter space encompassing all network weights and biases. At iteration $t$, the central server broadcasts the current global parameters $g_t$ to a participating subset of clients. Each participating client initializes its local model over $g_t$ and trains on its private dataset $D_i$, producing a local model $g_{\{t,i\}}$. The client then returns the update $\nabla_{\{t,i\}} = g_{\{t,i\}} - g_t$ to the central server. After that, the central server aggregates the collected client updates according to a predefined aggregation rule $F_{\text{AGR}}(\cdot)$, yielding the aggregated update $\nabla_t = F_{\text{AGR}}(\{\nabla_{\{t,i\}} | i \in N\})$. A new global model is subsequently updated as $g_{t+1} = g_t - \eta \nabla_t$, where $\eta$ is the global learning rate. This iterative procedure is repeated until the convergence criterion of global model is satisfied.

## 2.2 POISONING ATTACKS ON FL

Poisoning attacks in FL can generally be divided into two main types: data poisoning attacks and model poisoning attacks. Data poisoning attacks (Jagielski et al., 2018; Muñoz-González et al., 2017) involve adversaries contaminating the training datasets on their devices, while model poisoning attacks (Bagdasaryan et al., 2020; Baruch et al., 2019; Fang et al., 2020; Mhamdi et al., 2018; Xie et al., 2020) entail the direct manipulation of local model gradients by malicious participants, who then transmit these altered gradients to the server during the learning process. A notable study by Shejwalkar & Houmansadr (2021) mention two agnostic attacks in FL named Min-Max and Min-Sum. Min-Max aims to minimize the distance between benign and malicious clients, while Min-Sum looks for the minimized sum distances between benign and malicious clients. Without knowing the knowledge of the aggregation rules of the server, the attacks can compare the distances between themselves and benign clients instead, iteratively searching for an optimal parameter to update the malicious model, thereby achieving the performance of the attack. However, this strategy of looking for the minimum distances between benign and malicious is not realistic in the real-world FL scenarios. Additionally, the lack of control over the speed of the attack and the requirement for a significant proportion of malicious clients can lead to easy detection by robust AGRs, and low attack efficiency. Furthermore, once initiated, the predetermined attack objective in traditional attacks cannot be modified, which poses a limitation on the flexibility of the attack strategy.

## 2.3 EXISTING BYZANTINE-ROBUST DEFENSES

Current defenses against poisoning attacks in FL are categorized based on the detection and mitigation strategies servers employ to handle suspicious models. These strategies are generally grouped into three types: statistics-based, distance-based, and performance-based approaches (Shen et al., 2022). Statistics-based defenses, such as Median (Yin et al., 2021) and Trimmed Mean (Yin et al., 2021), use statistical features like mean or median to aggregate input gradients on each dimension, mitigating outlier impacts. Distance-based defenses, including methods like Krum (Blanchard et al., 2017), Mkrum (Blanchard et al., 2017), and Bulyan (Mhamdi et al., 2018), assess distances such as Euclidean distance or Cosine similarity between local updates to pinpoint statistical outliers. Performance-based defenses, exemplified by Fang (Fang et al., 2021), rely on a validation dataset to evaluate the performance of uploaded models, removing those that diverge significantly from expected outcomes. These varied approaches reflect the complexity of securing FL systems from sophisticated attacks aimed at compromising the collaborative learning process.

## 2.4 ADAPTIVE SLIDING MODE CONTROL

Adaptive Sliding Mode Control (ASMC) (Huang & Kuo, 2001) is a robust control technique that combines sliding mode control's insensitivity to matched disturbances and parameter uncertainties with adaptive mechanisms that can estimate unknown system parameters or disturbance bounds in real-time. The proposed approach, Foriers approximation technique (Huang & Kuo, 2001), systematically aggregates all uncertain parameters inherent in the controller synthesis process and represents these uncertainties through finite linear combinations of orthonormal basis functions. Consider a first order nonlinear system with the dynamic model $\dot{g}_t = u_t + d_t$, where $\dot{g}_t$ is the derivative of $g_t$ with respect to time $t$, and $d_t$ is the disturbance which is an unknown function of time. Conventionally, the sliding surface can be specified as:

$$\dot{s}_t = \dot{e}_t + \lambda e_t, \tag{2}$$

where $e_t = g_t - \tilde{g}$ is the loss function of the system state $g_t$ and the desired state $\tilde{g}$ at iteration $t$, and $\lambda$ is a hyperparameter to govern the convergence rate of $e_t$. If the controller can ensure that $s_t = 0$, i.e., $\dot{e}_t = -ke_t$, solving this first order differential equation will result in $e_t = e_0 e^{-kt}$, which converges exponentially to 0 as $t$ increases.

The next step establishes the design of control law $u_t$ to ensure $\lim_{t \to T} s_t = 0$ alongside real-time estimation of time-varying uncertainties $d_t$ to achieve desired tracking performance. The controller design follows a two-stage methodology: (i) formulation of control law $u_t$ to reach the sliding surface (Khoo et al., 2013; 2009; Alqumsan et al., 2019), and (ii) estimation of $d_t$ by finite-term Fourier series approximation technique (Huang & Kuo, 2001). The control law is chosen as:

$$u_t = -\lambda e_t - \hat{d}_t - \alpha \cdot \text{sign}(s_t), \tag{3}$$

where $\alpha$ is a positive constant selected to force the trajectory of the system to reach the sliding mode surface, and sign $(\cdot)$ is defined as follows: $\text{sign}(s_t) = \begin{cases} +1 & \text{if } s_t > 0, \\ 0 & \text{if } s_t = 0, . \\ -1 & \text{if } s_t < 0. \end{cases}$ $\hat{d}_t$ is the estimation of the unknown disturbance $d_t$, both of them can be approximated using finite-term Fourier series approximation as:

$$d_t = w_d^T \cdot z_t, \quad \hat{d} = \hat{w}_d^T \cdot z_t \tag{4}$$

where

$$w_d = [w_0, \ w_1, \ w_2, \ ... \ , w_{2nd}]^T \tag{5}$$

$$\hat{w}_d = [\hat{w}_0, \ \hat{w}_1, \ \hat{w}_2, \ ... \ , \hat{w}_{2nd}]^T \tag{6}$$

$$z_d = [1, \ \cos\omega_1 t, \ \sin\omega_1 t, \ \cos\omega_2, t \ \sin\omega_2 t, \ ... \ , \cos\omega_{nd} t, \ \sin\omega_{nd} t]^T \tag{7}$$

and the error is $\tilde{w}_d = w_d - \hat{w}_d$. By defining the energy function

$$V_t = \frac{1}{2} s_t^2 + \frac{1}{2} \tilde{w}_d Q_d \tilde{w}_d \tag{8}$$

where $Q_d \in \mathbb{R}^{(2nd+1) \times (2nd+1)}$ is a symmetric positive definite matrix.the convergence of the $s_t$ can be ensured when $\dot{V}_t \leq 0$ (i.e., $V_{t+1} < V_t$ for $V_t \neq 0$). Substituting Eq. 2 and Eq. 3 into $V_t$, we can get the time derivative of $V_t$ as:

$$\dot{V}_t = s_t[-\lambda e_t - \alpha \text{sign}(s_t) - \hat{d}_t + d_t + \lambda e_t] + \tilde{w}_d^T (z_d s - Q_d \dot{\hat{w}}_d) \tag{9}$$

Define the adaptive law $\hat{w}_d$ as

$$\dot{\hat{w}}_d = -Q_d^{-1} z_d s \tag{10}$$

then the time derivative of $V_t$ can be expressed as

$$\dot{V}_t \leq s_t(-\alpha \cdot \text{sign}(s_t)) = -\alpha|s_t| = -\sqrt{2}\alpha V_t^{1/2}. \tag{11}$$

Therefore, the convergence of the system can be guaranteed in finite-time (Yin et al., 2011; Khoo et al., 2013).

# 3 ASAP OVERVIEW

In this section, the detailed workflow of ASAP will be demonstrated. According to the example demonstrated in Sec. 2.4, the FL training process can be treated as a nonlinear system. Considering the practical implementation, we consider the attacker can only control those compromised clients for a limited number of communication rounds. In order to formulate the attack scenario, we firstly introduce the threat model of the attack.

## 3.1 THREAT MODEL

**Adversary's Goal** The goal of the adversary is to control the malicious clients updates therefore when the malicious gradients are uploaded to the central server, the accuracy of the global model can adaptively reduce to a target accuracy without the knowledge of AGRs.

**Adversary's Capability** We assume the adversary controls $m$ malicious clients of total $n$ clients, and $(m/n) < 0.5$. The agnostic adversary can access global parameters and directly manipulate the malicious clients gradients to the server. Moreover, we assume that the adversary does not know any knowledge of AGRs of central server or gradients of benign clients. In FL, malicious clients naturally have access to the global model.

**Comparing our attacks** LIE attacks (Baruch et al., 2019) estimate coordinate-wise mean and standard deviation of all client updates to generate statistically similar malicious perturbations. Min-Max and Min-Sum attacks (Shejwalkar & Houmansadr, 2021) constrain malicious updates within benign clusters using maximum distance or sum-of-distances bounds while pushing in adversarial

directions. In contrast, FMPA (Zhang et al., 2023) uses predictive reference models from histori-
cal data and subsequently fine-tunes them through gradient-based optimization to achieve desired
accuracy levels with precise control. However, as demonstrated in Fig. 1, our proposed attack funda-
mentally differs by seeking updates that are closest to the global optima rather than diverging from
it, thereby maintaining consistent effectiveness across different training phases and defensive mea-
sures without requiring statistical estimation, distance constraints, or iterative fine-tuning processes.

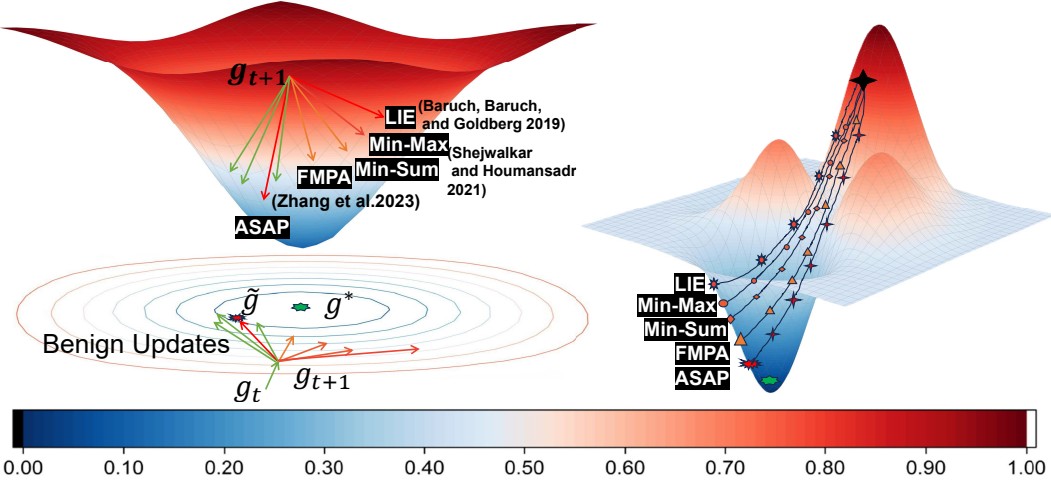

Figure 1: The comparison of existing attacks and our attack. ASAP can directly manipulate the
malicious model updates and then force the global model $g_{t+1}$ to reach the desired attack objective
$\tilde{g}$, which is chosen as the closest point to the global optima $g^*$. The attack effect is illustrated via
loss contours—blue area indicates low loss and red area indicates high loss.

## 3.2 ASAP'S ALGORITHM

We treat the overall FL global model

$$g_t = F_{\text{AGR}}\{g_{\{t,1\}}, g_{\{t,2\}}, ..., g_{\{t,m\}}, ..., g_{\{t,N\}}\} \tag{12}$$

as a nonlinear system, and in particular, the malicious local models are chosen as

$$\dot{g}'_{\{t,i \in m\}} = u_t. \tag{13}$$

The goal of ASAP is to design the control law $u_t$, and design the adaptive law $\hat{w}_\Phi$ by applying
the function approximation technique using Fourier Series to transform the uncertainties into a finite
combination of orthonormal basis functions—thus to ensure that the global model $g_t$ will slide along
the surface $s_t = 0$ to achieve:

$$e_t(\tilde{g}, g_t) = -C/k \tag{14}$$

exponentially fast, where $C \in \mathbb{R}$ is a constant to adjust the convergence status of $e_t$, and $k \in \mathbb{R}$
($k \neq 0$) is a parameter to adjust the convergence speed of $e_t$. To achieve the adversary's goal, we
design the error function as

$$e_t = g_t - \tilde{g}, \tag{15}$$

To realize this new error, we design the sliding surface as

$$s_t = \int (\dot{e}_t + k e_t + C)\text{d}t + C_1, \tag{16}$$

where $C_1 \in \mathbb{R}$ is the initial value of the sliding surface $s_t$, which can be any constant, and $\Phi_t$ is the
unknown disturbance.

After selecting the sliding surface $s_t$, the control law $u_t$ is designed based on the FL system, the
dynamic model in Eq. 13, the error function in Eq. 15 and sliding surface $s_t$ in Eq. 16, as follows:

$$u_t = \left[\frac{dg_t}{dg'_{\{t,i\}}}\right]^{-1}[-k e_t - \eta \text{sign}(s_t) - \hat{\Phi}_t - C], \tag{17}$$

where $\eta > 0$ is a positive constant selected to force the system trajectory to reach the sliding mode surface. Here, $dg_t/dg'_{\{t,i\}}$ is the derivative of $g_t$ with respect to $g'_{\{t,i\}}$ and $\hat{\Phi}_t$ is the estimation function of $\Phi_t$. Using the Fourier Series and approximation technique to estimate $\Phi_t$, it can be represented as:

$$\Phi_t = w_\Phi^T z_\Phi, \quad \hat{\Phi}_t = \hat{w}_\Phi^T z_\Phi \tag{18}$$

where

$$w_\Phi = [w_0, \ w_1, \ w_2, \ \ldots, w_{2n\Phi}]^T \tag{19}$$

$$\hat{w}_\Phi = [\hat{w}_0, \ \hat{w}_1, \ \hat{w}_2, \ \ldots, \hat{w}_{2n\Phi}]^T \tag{20}$$

$$z_\Phi = [1, \ \cos\omega_1 t, \ \sin\omega_1 t, \ \cos\omega_2 t, \ \sin\omega_2 t, \ \ldots, \cos\omega_{n\Phi} t, \ \sin\omega_{n\Phi} t]^T \tag{21}$$

$w_\Phi \in \mathbb{R}^{2n\Phi+1}$ is the weighting parameter, $\hat{w}_\Phi \in \mathbb{R}^{2n\Phi+1}$ is the estimated weighting parameter, and $z_\Phi \in \mathbb{R}^{2n\Phi+1}$ is the vector of orthonormal basis function. Note that the number of $n_\Phi$ needs to be chosen bigger enough to ensure the performance of approximation of $\Phi_t$. The adaptive law of $w_\Phi$ is defined as:

$$\dot{\hat{w}}_\Phi = -Q_\Phi^{-1} z_\Phi s \tag{22}$$

where $Q_\Phi \in \mathbb{R}^{(2n\Phi+1)\times(2n\Phi+1)}$ is a symmetric positive definite matrix. The workflow of ASAP to compromise a client is demonstrated in Algorithm 1 in Appendix A.2.

### 3.3 ASAP Convergence Analysis

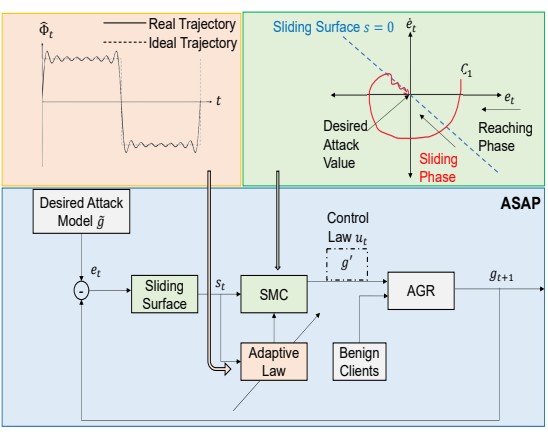

Figure 2: The block diagram of ASAP. The function of adaptive law is to automatically adjust the weight of the estimator in order to track the unknown function. In the SMC block, the system state is forced to slide along the sliding surface which means global model is forced to the desired attack objectives.

The convergence analysis is illustrated in Theorem 3.1 as below.

**Theorem 3.1.** *Consider a FL system characterized by the dynamics in Eq. 13, with error function specified in Eq. 15 and a sliding manifold defined by Eq. 16. Given the control law $u_t$ formulated in Eq. 17 with parameters $k > 0$, $\eta > 0$, $C \in \mathbb{R}$, and the derivative of the aggregation function $F_{AGR}$ with respect to the malicious model $g'_{\{t,i\}}$ is continuous. Then the ASMC framework guarantees: (i) Fourier series approximation of the unknown uncertainty $\Phi_t$; (ii) finite-time convergence of the sliding surface $s_t$ to zero with subsequent invariance; (iii) exponential convergence of the error $e_t = g_t - \tilde{g}$ to $-C/k$.*

*Note that the theoretical proof provided addresses scalar dynamics rather than vector dynamics. Since a vector is composed of multiple scalars, proving the property for each individual scalar inherently establishes the same property for the entire vector. Thus, demonstrating the desired property at the scalar level is sufficient to confirm the corresponding property for the vector as a whole.*

Due to space limit, the proof of Theorem 3.1 is delayed to Appendix A.1. Below, we highlight significant remarks on the new features of ASAP.

**Remark 1: AGR-Agnostic Operation.** Unlike existing AGR-agnostic attacks (LIE, Min-Max, Min-Sum) that still require statistical estimation of benign client updates, ASAP achieves complete independence from both aggregation rules and benign client information. The ASMC framework treats unknown aggregation effects as system disturbances $\Phi_t$, which are estimated in real-time through Fourier series approximation without requiring any prior knowledge of $F_{AGR}$ or benign gradient statistics.

**Remark 2: Convergence Speed.** The parameter $k$ serves as a convergence rate controller, enabling precise manipulation of $e_t$. On the sliding surface where $s_t = \dot{s}_t = 0$, solving the differential

Table 1: The comparison of the accuracy of the global model between different attacks on CIFAR10, MNIST, and Tiny ImageNet against different AGRs. More experimental results against different AGRs under various attack objectives are demonstrated in Appendix A.3.4.

| Dataset (Model) | AGRs | No Attack(%) | Test Acc. (Difference to the Targeted Acc. $\varsigma$ (%)) | | | | |
| --- | --- | --- | --- | --- | --- | --- | --- |
| | | | LIE | Min-Max | Min-Sum | FMPA | ASAP |
| | | | Target Acc 60% | | | | |
| CIFAR10 (AlexNet) | FedAvg | 66.42 | 53.28 (-11.20) | 32.75 (-45.42) | 51.06 (-14.90) | 64.33 (7.22) | 61.58 (**2.63**) |
| | Median | 64.28 | 33.40 (-44.33) | 28.08 (-53.20) | 33.73 (-43.78) | 63.57 (5.95) | 56.53 (**-5.78**) |
| | Trmean | 66.23 | 46.43 (-23.78) | 30.95 (-48.42) | 41.19 (-31.52) | 55.44 (-7.60) | 61.87 (**3.12**) |
| | NB | 66.73 | 51.95 (-13.42) | 45.64 (-24.07) | 55.51 (-7.48) | 64.29 (7.15) | 61.33 (**2.22**) |
| | Bulyan | 66.07 | 36.91 (-38.48) | 25.95 (-56.75) | 23.52 (-60.80) | 62.55 (4.25) | 61.87 (**3.12**) |
| | Mkrum | 66.79 | 45.03 (-24.95) | 52.29 (-12.85) | 31.74 (-47.11) | 63.26 (5.43) | 60.65 (**0.92**) |
| | Fltrust | 66.59 | 31.53 (-47.42) | 50.79 (-15.18) | 52.56 (-12.40) | 65.52 (9.20) | 61.94 (**3.23**) |
| | CC | 66.62 | 63.53 (5.88) | 10.53 (-82.45) | 14.94 (-75.10) | 67.22 (12.03) | 62.13 (**3.55**) |
| | DNC | 66.55 | 62.92 (4.87) | 63.94 (6.57) | 58.26 (-2.90) | 65.01 (8.35) | 61.25 (**2.08**) |
| | | | Target Acc 10% | | | | |
| | FedAvg | 66.42 | 53.28 (432.80) | 32.75 (227.50) | 51.06 (410.60) | 19.83 (98.30) | 10.73 (**7.30**) |
| | Median | 64.28 | 33.40 (234.00) | 28.08 (180.80) | 33.73 (237.30) | 13.52 (35.20) | 10.65 (**6.50**) |
| | Trmean | 66.23 | 46.43 (364.30) | 30.95 (209.50) | 41.19 (311.90) | 18.94 (89.40) | 9.98 (**-0.20**) |
| | NB | 66.73 | 51.95 (419.50) | 45.64 (356.40) | 55.51 (455.10) | 20.35 (103.50) | 10.08 (**0.80**) |
| | Bulyan | 66.07 | 36.91 (269.10) | 25.95 (159.50) | 23.52 (135.20) | 19.76 (97.60) | 9.95 (**-0.50**) |
| | Mkrum | 66.79 | 45.03 (350.30) | 52.29 (422.90) | 31.74 (217.40) | 16.58 (65.80) | 10.72 (**7.20**) |
| | Fltrust | 66.59 | 31.53 (215.30) | 50.79 (407.90) | 52.56 (425.60) | 25.89 (158.90) | 10.94 (**9.40**) |
| | CC | 66.62 | 63.53 (535.30) | 10.53 (5.30) | 14.94 (49.40) | 26.17 (161.70) | 10.48 (**4.80**) |
| | DNC | 66.55 | 62.92 (529.20) | 63.94 (539.40) | 58.26 (482.60) | 14.73 (47.30) | 10.76 (**7.60**) |
| | | | Target Acc 90% | | | | |
| MNIST (MLP) | FedAvg | 97.98 | 94.12 (4.58) | 91.67 (1.85) | 92.84 (3.16) | 95.28 (5.87) | 91.04 (**1.16**) |
| | Median | 97.81 | 90.99 (1.10) | 91.15 (1.28) | 92.84 (3.16) | 43.79 (-51.34) | 88.22 (**-1.98**) |
| | Trmean | 97.42 | 91.80 (2.00) | 91.30 (1.44) | 92.43 (2.70) | 97.26 (8.07) | 90.69 (**0.77**) |
| | NB | 97.96 | 92.82 (3.13) | 91.88 (2.09) | 93.02 (3.36) | 60.20 (-33.11) | 90.95 (**1.06**) |
| | Bulyan | 97.97 | 88.92 (-1.20) | 91.96 (2.18) | 92.29 (2.54) | 45.28 (-49.69) | 89.22 (**-0.87**) |
| | Mkrum | 97.94 | 92.33 (2.59) | 96.14 (6.82) | 95.39 (5.99) | 93.41 (3.79) | 92.19 (**2.43**) |
| | Fltrust | 97.96 | 87.89 (-2.34) | 73.49 (-18.34) | 93.12 (3.47) | 95.01 (5.57) | 92.46 (**2.73**) |
| | CC | 97.96 | 95.35 (5.94) | 94.61 (5.12) | 94.54 (5.04) | 96.99 (7.77) | 93.54 (**3.93**) |
| | DNC | 97.95 | 93.08 (3.42) | 92.90 (3.22) | 93.36 (3.73) | 93.22 (3.58) | 92.46 (**2.73**) |
| | | | Target Acc 10% | | | | |
| | FedAvg | 97.98 | 94.12 (841.20) | 91.67 (816.70) | 92.84 (828.40) | 12.18 (21.80) | 10.47 (**4.70**) |
| | Median | 97.81 | 90.99 (809.90) | 91.15 (811.50) | 92.84 (828.40) | 11.92 (19.20) | 10.79 (**7.90**) |
| | Trmean | 97.42 | 91.80 (818.00) | 91.30 (813.00) | 92.43 (824.30) | 16.34 (63.40) | 10.06 (**0.60**) |
| | NB | 97.96 | 92.82 (828.20) | 91.88 (818.80) | 93.02 (830.20) | 15.27 (52.70) | 10.34 (**3.40**) |
| | Bulyan | 97.97 | 88.92 (789.20) | 91.96 (819.60) | 92.29 (822.90) | 12.06 (20.60) | 10.25 (**2.50**) |
| | Mkrum | 97.94 | 92.33 (823.30) | 96.14 (861.40) | 95.39 (853.90) | 15.81 (58.10) | 10.83 (**8.30**) |
| | Fltrust | 97.96 | 87.89 (778.90) | 73.49 (634.90) | 93.12 (831.20) | 35.47 (254.70) | 10.12 (**1.20**) |
| | CC | 97.96 | 95.35 (853.50) | 94.61 (846.10) | 94.54 (845.40) | 44.28 (342.80) | 10.86 (**8.60**) |
| | DNC | 97.95 | 93.08 (830.80) | 92.90 (829.00) | 93.36 (833.60) | 30.64 (206.40) | 10.39 (**3.90**) |
| | | | Target Acc 45% | | | | |
| Tiny ImageNet (ResNet50) | FedAvg | 57.49 | 51.63 (14.73) | 38.37 (-14.73) | 53.20 (18.22) | 54.64 (21.42) | 48.07 (**6.82**) |
| | Median | 53.47 | 22.14 (-50.80) | 54.08 (20.18) | 34.24 (-23.91) | 42.94 (-4.58) | 46.93 (**4.29**) |
| | Trmean | 54.78 | 51.60 (14.67) | 54.59 (21.31) | 39.82 (-11.51) | 55.90 (24.22) | 44.94 (**-0.13**) |
| | NB | 58.62 | 52.98 (17.73) | 52.95 (17.67) | 53.09 (18.20) | 56.12 (24.71) | 45.57 (**1.27**) |
| | Bulyan | 54.93 | 24.93 (-44.60) | 48.01 (6.69) | 33.51 (-25.53) | 5.15 (-88.56) | 44.98 (**-0.01**) |
| | Mkrum | 54.96 | 27.02 (-39.96) | 49.68 (10.40) | 26.39 (-41.36) | 36.06 (-19.87) | 45.46 (**1.02**) |
| | Fltrust | 54.35 | 33.57 (-25.40) | 47.04 (4.53) | 53.45 (18.78) | 55.48 (23.29) | 45.31 (**0.69**) |
| | CC | 54.31 | 29.13 (-35.27) | 32.26 (-28.31) | 30.99 (-31.13) | 47.88 (6.40) | 44.13 (**-1.93**) |
| | DNC | 55.97 | 68.12 (51.38) | 69.66 (54.80) | 54.29 (20.64) | 46.98 (4.40) | 44.36 (**-1.42**) |
| | | | Target Acc 0.5% | | | | |
| | FedAvg | 57.49 | 51.63 (10226.00) | 38.37 (7574.00) | 53.20 (10540.00) | 52.31 (10362.00) | 0.53 (**6.00**) |
| | Median | 53.47 | 22.14 (4328.00) | 54.08 (10716.00) | 34.24 (6748.00) | 51.48 (10196.00) | 0.51 (**2.00**) |
| | Trmean | 54.78 | 51.60 (10220.00) | 54.59 (10818.00) | 39.82 (7864.00) | 48.27 (9554.00) | 0.53 (**6.00**) |
| | NB | 58.62 | 52.98 (10496.00) | 52.95 (10490.00) | 53.09 (10518.00) | 50.76 (10052.00) | 0.52 (**4.00**) |
| | Bulyan | 54.93 | 24.93 (4886.00) | 48.01 (9502.00) | 33.51 (6602.00) | 18.35 (3570.00) | 0.52 (**4.00**) |
| | Mkrum | 54.96 | 27.02 (5304.00) | 49.68 (9836.00) | 26.39 (5178.00) | 42.14 (8328.00) | 0.53 (**6.00**) |
| | Fltrust | 54.35 | 33.57 (6614.00) | 47.04 (9308.00) | 53.45 (10590.00) | 49.83 (9866.00) | 0.52 (**4.00**) |
| | CC | 54.31 | 29.13 (5726.00) | 32.26 (6352.00) | 30.99 (6098.00) | 41.65 (8230.00) | 0.51 (**2.00**) |
| | DNC | 55.97 | 68.12 (13524.00) | 69.66 (13832.00) | 54.29 (10758.00) | 43.89 (8678.00) | 0.51 (**2.00**) |

equation $\dot{e}_t = -ke_t - C$, produces $e_t = 1/k \cdot e_0^{-kt} - C/k$. The analytical solution reveals that $k$ determines the exponential convergence characteristics: larger values of $k$ correspond to faster exponential convergence rates. This mathematical property enables ASAP to offer flexible convergence speed modulation capabilities.

**Remark 3: Adjustable Objectives.** The adversary can dynamically modify attack objectives throughout ASAP execution by appropriately selecting parameter $C$ in $e_t$ as evaluated in Eq. 16. When the system reaches equilibrium on the sliding manifold where both $\dot{s}_t = 0$ and $\dot{e}_t = 0$, the constraint $\dot{s}_t = \dot{e}_t + ke_t + C$ results in the equilibrium relationship $e_t = -C/k$ or $g_t = \tilde{g} + C/k$.

**Continuous-time formulation**   The fundamental reason for using continuous time analysis is because structured mathematical rules, especially differentiation, chain rules, and so on are all well established by mathematicians through measure theory. It is therefore valid to analyze the system in continuous time. Similar to the analysis of back-propagation and steepest descent algorithms for example, when describing back propagation training algorithm, continuous time analysis is used.

Regarding of Eq. 13, the dynamic model is used to analyze how the malicious clients update over time, not implement. Based on the definition of derivative, the derivative of malicious model at

$t$, $g'_{\{t,i\}}$ can be represented as $\dot{g}'_{\{t,i\}} = \lim_{\Delta t \to 0} \frac{g'_{(\{t+\Delta t),i\}} - g'_{\{t,i\}}}{\Delta t}$. Therefore, in practical attack scenarios, the rate of change of the model, $\dot{g}'_{\{t,i\}}$, can be approximated by the difference in values divided by a small time interval, effectively capturing the derivative's behavior in discrete time.

**Non-differentiable AGRs**   ASAP does not require differentiability of the aggregation rule itself. In particular, we never differentiate through the AGR (e.g., Median, Krum), but only through the global model $g_t$ and the malicious model $g'_{\{t,i\}}$, both instantiated as neural networks (AlexNet, MLP, ResNet50) in our experiments. In the federated learning pipeline, even if the AGR $F_{\text{AGR}}$ is non-smooth, its output $g_t = F_{\text{AGR}}\{g_{\{t,i\}}\}$ is simply a collection of neural network parameters. All derivatives involved in the control law $u_t$ in Eq. 17 and in the proof of Theorem 3.1 (Appendix A.1) are taken with respect to these model parameters, i.e., $g_t$ and $g'_{t,i}$, rather than with respect to the AGR mapping or any coordinate-wise non-smooth statistics. Consequently, no differentiation in our analysis involves non-smooth functions of the parameters.

# 4   PERFORMANCE EVALUATION

## 4.1   EXPERIMENT SETTINGS

**Datasets and Models**   Our experimental evaluation of ASAP encompasses diverse architectures and benchmark datasets. We deploy AlexNet following Yang (Yang et al., 2017) for CIFAR10 experiments, utilize a fully connected (FC) neural network architecture for MNIST (Deng, 2012), and employ ResNet50 for Tiny ImageNet (Le & Yang, 2015) evaluation. The experimental framework incorporates both Independent and Identically Distributed (IID) and Non-Independent and Identically Distributed (Non-IID) data partitioning schemes. For Non-IID configurations, we leverage the Dirichlet distribution parameterized by concentration values $\{0.1, 0.3, 0.5, 0.7, 0.9\}$ to systematically vary data heterogeneity levels. Smaller concentration parameters (e.g., 0.1) generate severely imbalanced client datasets with pronounced class skewness, while larger values approach uniform class distributions across participating clients. The experimental configurations are tailored to optimize performance across different architecture-dataset combinations. Comprehensive details of each dataset are provided in Appendix A.3.2.

**Attack Settings**   The experimental setup involves a federated network of 50 clients with a 10% malicious participation rate, consistent with established benchmarks in adversarial federated learning research (Zhang et al., 2023; Shejwalkar & Houmansadr, 2021; Baruch et al., 2019). Under our threat model, adversaries gain control over compromised client devices, enabling strategic manipulation of local parameter updates to achieve precise global model accuracy targets. The attack targets are stratified across datasets: CIFAR10 targets at 60% (reference), 55%, 50%, and 10% through $C$ parameter tuning. MNIST configurations target 90% (reference), 85%, 80%, and 10% accuracies via $C$ adjustment. Tiny ImageNet targets of 45% (reference), 40%, 35%, and 0.5% through $C$ modulation. The lower bounds (10% for CIFAR10/MNIST, 0.5% for Tiny ImageNet) represent random guess performance baselines. We compare our attack with existing methods including AGR-agnostic approaches LIE (Baruch et al., 2019), Min-Max (Shejwalkar & Houmansadr, 2021), and Min-Sum (Shejwalkar & Houmansadr, 2021), as well as FMPA (Zhang et al., 2023) which provides precise control capabilities but requires AGR knowledge. The details of each attack are introduced in Appendix A.3.2.

**Evaluation Defenses**   In the experiments, various defenses are considered such as FedAvg (McMahan et al., 2017), Median (Yin et al., 2021), Trmean (Yin et al., 2021), Norm-Bounding (NB) (Sun et al., 2019) Bulyan (Mhamdi et al., 2018), Mkrum (Blanchard et al., 2017), Fltrust (Cao et al., 2022), CC (Karimireddy et al., 2021), and DNC (Shejwalkar & Houmansadr, 2021). The details of each defense are demonstrated in Appendix A.3.3.

**Evaluation Metric**   Define $I_T$ and $I_0$ as the target and achieved attack accuracies, respectively. The normalized deviation $\varsigma = ((I_T - I_0)/I_T) \times 100\%$ measures the relative distance between attack objectives and actual results. Attack method comparison employs the absolute metric $|\varsigma|$, where smaller values denote better objective fulfillment and higher attack quality.

## 4.2 EXPERIMENTS RESULTS

Experimental results presented in Table 1 and Figure 3 demonstrate the comparative performance of attack methods against different AGRs using CIFAR10/AlexNet, MNIST/MLP, and Tiny ImageNet/ResNet50 benchmarks. More experimental results under different scenarios are demonstrated in Appendix A.3.4. Overall, ASAP achieves the minimal $|\varsigma|$ values and consistently outperforms all baseline attacks.

As shown in Figure 3, ASAP achieves robust convergence to attack objectives without triggering AGR detection, requiring fewer communication rounds than competing methods. In contrast to AGR-agnostic attacks including LIE, Min-Max and Min-Sum, which fail to achieve precise control and demand increased communication resources, and unlike FMPA, which encounters detection by AGRs under various conditions, causing the test accuracy to converge near the optimal performance achieved without any attack presence.

Table 2: Time Complexity and Effective Communication Rounds comparisons.

| Comparison | LIE | Min-Max | Min-Sum | FMPA | ASAP |
|---|---|---|---|---|---|
| Time (hrs) | 0.8 | 0.9 | 0.9 | 1.0 | 1.1 |
| Rounds (epochs) | 781 | 778 | 767 | 34 | 19 |

The comprehensive evaluation demonstrates ASAP's consistent performance compared to existing SOTA AGR-agnostic attack methods, coupled with fine-grained controllability for precise attack execution. The subsequent discussion examines the findings across three key dimensions.

**Time Complexity**  The computational cost analysis, detailed in Table 2, reveals that ASAP demonstrates the highest execution time among evaluated methods, primarily due to the computational demands of its underlying mathematical framework. Nevertheless, the increased computational cost compared to competing methods remains feasible for practical deployment.

**Effective Communication Rounds**  To maintain evaluation consistency, we utilize Effective Communication Rounds (ECR) as the standardized communication efficiency metric. Table 2 presents the average convergence performance on CIFAR10 dataset, establishing that ASAP requires the minimum number of communication rounds to achieve attack objectives compared to existing approaches.

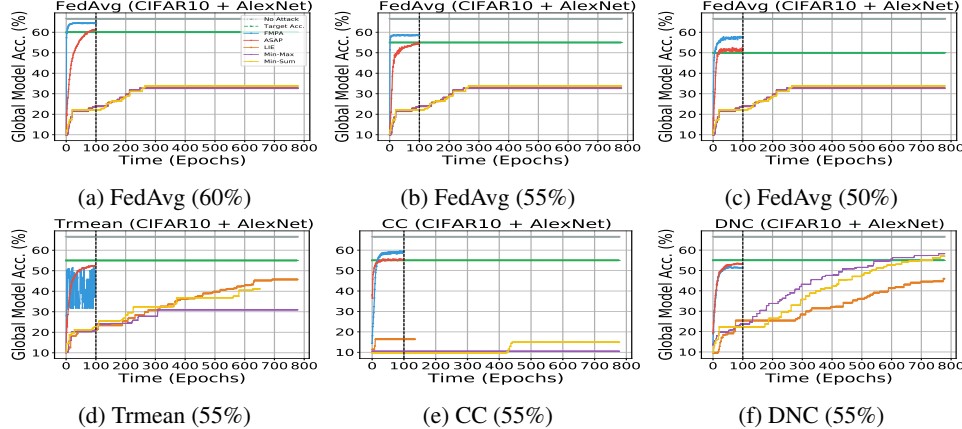

Figure 3: Comparison of each attack against various AGRs with different attack objectives on CIFAR10 with AlexNet under different attack objectives and different attacks under the same target accuracy. Comparison figures on MNIST and Tiny ImageNet are given in Appendix A.3.4.

**Precise Control**  Table 3 in Appendix A.3.4 presents comprehensive evaluation results across multiple AGRs under diverse attack objectives. ASAP consistently exhibits the lowest $|\varsigma|$ scores while surpassing all comparative attacks, validating its capability for accurate objective targeting with minimal loss variance. The CIFAR10 results show average $|\varsigma|$ values of 2.18%, 2.61%, and 1.62% for attack objectives of 60%, 55%, and 50% respectively.

### 4.2.1 ABLATION STUDY

**Impact of percentage of attackers**   The impact of malicious client proportion on FL is analyzed by incrementally increasing the adversarial ratio from 5% to 20%. As illustrated in Figure 4a, ASAP exhibits consistent performance advantages compared to competing attack strategies across all evaluated ratios.

**Impact of Non-IID degrees**   The impact of data heterogeneity on attack efficacy is assessed using CIFAR10 with Dirichlet concentration parameters spanning {0.1, 0.3, 0.5, 0.7, 0.9}, targeting 55% accuracy under Trmean aggregation. As presented in Figure 4b, ASAP successfully accomplishes the attack objectives while demonstrating robust outperform of existing attack strategies regardless of statistical heterogeneity intensity.

**Impact of number of clients**   While our baseline experiments employ 50 total clients, we extend the evaluation to assess ASAP's scalability under larger federation sizes of 100, 150, and 200 participants using CIFAR10 with a 55% target accuracy. Figure 4c demonstrates that ASAP maintains consistent superiority over competing attack methods across all federation scales.

**Impact of clients sampling rates**   The impact of client sampling rate variations on attack performance is examined in Figure 4d. Experimental findings indicate that ASAP exhibits enhanced consistency and reduced performance variance relative to competing attack approaches across all sampling configurations.

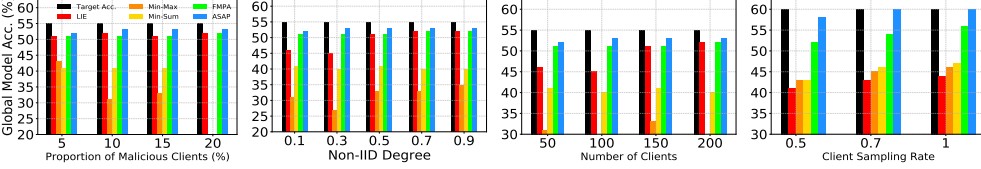

(a) Different proportions of malicious clients
(b) Different degrees of Non-IID
(c) Different number of clients
(d) Different sampling rates of clients

Figure 4: Ablation study results against Trmean on CIFAR10 with AlexNet to target accuracy at 55%.

## 5 CONCLUSION

In this paper, we introduced ASAP, a novel AGR-agnostic model poisoning attack on Federated Learning, inspired by Adaptive Sliding Mode Control theory. Unlike prior agnostic attacks that rely on heuristic distance-based strategies or require partial knowledge of benign updates, ASAP formulates the poisoning process as a controllable nonlinear system. By leveraging a Fourier series-based estimator, ASAP precisely tracks the global model trajectory and adaptively adjusts the direction and magnitude of malicious updates toward a predefined target. This enables both fine-grained control over convergence speed and resilience against diverse aggregation rules.

Our theoretical analysis guarantees convergence to the attack objective under finite time, without requiring knowledge of the server's aggregation strategy or benign client behavior. Extensive experiments on CIFAR-10, MNIST, and Tiny ImageNet across various robust AGRs which demonstrate that ASAP consistently outperforms SOTA AGR-agnstic attacks in both convergence efficiency and target alignment.

ASAP opens a new attack surface in FL by enabling precise, stealthy, and adaptive poisoning. To counteract this threat, future research should explore dynamic defense mechanisms. In particular, we propose leveraging system identification techniques to model and detect abnormal update dynamics introduced by adaptive attackers. By identifying deviations from expected system behavior, such defenses could adaptively reject suspicious updates in real time.

ETHICS STATEMENT

This work adheres to the ICLR Code of Ethics. In this study, no human subjects or animal experimentation was involved. All datasets used, including CIFAR10, MNIST and Tiny ImageNet, were sourced in compliance with relevant usage guidelines, ensuring no violation of privacy. We have taken care to avoid any biases or discriminatory outcomes in our research process. No personally identifiable information was used, and no experiments were conducted that could raise privacy or security concerns. We are committed to maintaining transparency and integrity throughout the research process.

REPRODUCIBILITY STATEMENT

We have made every effort to ensure that the results presented in this paper are reproducible. All code and datasets have been made publicly available in an anonymous repository to facilitate replication and verification at `https://github.com/ICLR2026-ASAP/ASAP`. The experimental setup, including training steps, model configurations, and experimental setting details, is described in detail in the paper.

Additionally, the public benchmark datasets used in this paper, such as CIFAR10, MNIST and Tiny ImageNet, are publicly available, ensuring consistent and reproducible evaluation results.

We believe these measures will enable other researchers to reproduce our work and further advance the field.

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

# A APPENDIX

## A.1 PROOF OF THEOREM 3.1

*Proof.* To design the update law for $\hat{w}_\Phi$, defining $\tilde{w}_\Phi = w_\Phi - \hat{w}_\Phi$ and a Lyapunov function (or energy function) as below:

$$V_t = \frac{1}{2}s_t^2 + \frac{1}{2}\tilde{w}_\Phi Q_\Phi \tilde{w}_\Phi \tag{23}$$

where $Q_\Phi \in \mathbb{R}^{(2n\Phi+1)\times(2n\Phi+1)}$ is a symmetric positive definite matrix. After differentiating $V_t$ with respect to time, we have

$$\dot{V}_t = s_t\dot{s}_t + \tilde{w}_\Phi^T Q_\Phi \tilde{w}_\Phi \tag{24}$$

$$= s_t(\frac{dg_t}{dg'_{\{t,i\}}}\dot{g}_{\{t,i\}} + \Phi_t + ke_t + C) - \tilde{w}_\Phi^T Q_\Phi \dot{\hat{w}}_\Phi. \tag{25}$$

Using control law

$$u_t = \left[ \frac{dg_t}{dg'_{\{t,i\}}} \right]^{-1} [-ke_t - \eta\text{sign}(s_t) - \hat{\Phi}_t - C], \tag{26}$$

and the adaptive law

$$\dot{\hat{w}}_\Phi = -Q_\Phi^{-1} z_\Phi s \tag{27}$$

we get

$$\dot{V}_t = s_t[-ke_t - \eta\text{sign}(s_t) - C - \hat{\Phi}_t + \Phi_t + ke_t + C]$$
$$+ \tilde{w}_\Phi^T(z_\Phi s - Q_\Phi \dot{\hat{w}}_\Phi) \tag{28}$$
$$\leq s_t[-\eta_1\text{sign}(s_t)] \tag{29}$$
$$= -\eta_1|s_t| = -\sqrt{2}\eta_1 V_t^{1/2} \tag{30}$$

where $\eta = \eta_1 + \delta$, $\delta > 0$. By the finite time stability theorem proved in the study (Khoo et al., 2009), $V_t$ will converge to zero in a finite time, and hence results in $s_t = \dot{s}_t = 0$ in a finite time. □

## A.2 ALGORITHM

In this section, the algorithm of ASAP is demonstrated. We firstly initialize the value of $g_t$, $s_t$ as $g_0$, $s_0$ respectively (line 2), and malicious clients need to initialize weight of the estimator $\hat{w}_\Phi$. At the $t$-th communication round, the client is selected by the server and receives the current global model $g_t$.

---

**Algorithm 1** The workflow of ASAP to compromise a client

---

**Require:** Global model $g_t$, desired poisoning model $\tilde{g}$.
**Ensure:** malicious model update $g'_t$.
 1: **if** t=0 **then**
 2:     $g_t \leftarrow g_0, \tilde{g}$
 3:     Initialize $\hat{w}_\Phi$
 4: **else**
 5:     **for** malicious client $i = 1$ to $m$ **do**
 6:         Update the adaptive law in Eq. 22
 7:         Calculate $\hat{\Phi}_t$ in Eq. 18
 8:         Calculate $e_t$ of $g_t$ and $g_t^*$ in Eq. 15
 9:         Calculate $s_t$ in Eq. 16
10:     **end for**
11:     calculate $g'_t$ from Eq. 17                              ▷ {control law}
12:     **Output** $g'_t$
13: **end if**
14: Update the malicious client model $g'_t$ on FL

---

## A.3 DATASETS, ATTACKS AND DEFENSES

In this section, we give details of our experiments settings. For CIFAR10 experiments with AlexNet, we establish a global learning rate of 0.02, a global batch size of 128, and conduct training over 100 global rounds, with local client updates using a batch size 10 across 5 local epochs. MNIST experiments employing MLP utilize a global learning rate of 0.01, a global batch size of 128, and 100 training rounds, while local training proceeds with a batch size 5 over 3 epochs. The Tiny ImageNet-ResNet50 configuration employs a global learning rate of 0.001, maintains a batch size 128, and executes 20 global rounds, with local updates using a batch size 10 for 3 epochs. These hyperparameter selections reflect architecture-specific optimization requirements and dataset complexity considerations.

### A.3.1 Datasets

- **CIFAR10** (Krizhevsky, 2009). It is an image database with 60,000 colour images of 32 * 32 size in 10 classes equally, and it is divided into training dataset with 50,000 images and test dataset with 10,000 images.

- **MNIST** (Deng, 2012). It is a dataset with 70,000 hand-written digital images in 28 * 28 size with 10 classes equally, and it is divided into training dataset with 60,000 images and test dataset with 10,000 images.

- **Tiny ImageNet** (Le & Yang, 2015) It is a subset of ILSVRC (ImageNet challenge) (Deng et al., 2009), which is one of the most famous benchmarks for image classification. As a subset, Tiny ImageNet only has 200 different classes. In addition, each class contains 500 training images, 50 validation images, and 50 test images totally. Moreover, the size of the images is revised to 64 * 64 pixels instead of 224 * 224 pixels in standard ImageNet.

### A.3.2 Attacks

- **LIE** (Baruch et al., 2019). It inserts an appropriate amounts of noise which are large for the adversary to impact the global model while small to avoid attention by Byzantine-robust AGRs to each dimension of the average of the benign gradients.

- **Min-Max** (Shejwalkar & Houmansadr, 2021). They minimize the distance of malicious clients to benign clients, and then ensure the poisoned updates lie closely to the clique of benign gradients.

- **Min-Sum** (Shejwalkar & Houmansadr, 2021). They minimize the the sum of the squared distance of malicious clients to benign clients, and then ensure the poisoned updates lie closely to the clique of benign gradients.

- **FMPA** (Zhang et al., 2023). It generates an estimator to predict the global model in the next iteration as a benign reference model to fine-turn the global model to the desired poisoned model by collecting the historical information.

### A.3.3 Defenses

- **FedAvg** (McMahan et al., 2017). It is a basic algorithm on FL without defense. It collects all the local updates from the clients and computes the average of them as the output of aggregation.

- **Median** (Yin et al., 2021). It computes the median of the values from each dimension of gradients as a new global gradient.

- **Trmean (Trimmed-mean)** (Yin et al., 2021). It drops the specific number of maximum and minimum values from the local updates from the clients, and use the average value of the remaining updates as the aggregation output.

- **Norm-bounding** (Sun et al., 2019). It will scale the local update of the clients if the $l_2$ norm of it is bigger than the fixed threshold. Then it will average the scaled local updates as it's aggregation.

- **Bulyan** (Mhamdi et al., 2018). It uses Mkrum to select the updates as a selection set and then use Trmean (Yin et al., 2021) to aggregate the gradients. Trmean averages the gradients after removing the $m$ largest and smallest values from the updates, $m$ is usually set as the number of malicious clients.

- **Mkrum** (Blanchard et al., 2017). It was modified by krum (Blanchard et al., 2017) to aggregate the information provided from the clients effectively. Krum selects the single gradient which is closest to $(N - m - 2)$ neighboring gradients, where $N$ and $m$ are the number of all clients and malicious clients respectively. Mkrum selects multi gradients using krum to obtain a selection set and then average the gradients.

- **Fltrust** (Cao et al., 2022). It assigns a trust score to each clients based on the updates from them to the global update direction, the lower trust score the client get, the more the direction deviates. Then Fltrust normalizes the gradients of local model updates by the trust cores, and then average the updates as a global model.

- **CC (Centered Clipping)** (Karimireddy et al., 2021). It clips all the gradients to the bad vector $\rho$ to ensure the error is less than a specific value. Then it averages the normalized local updates with the weight of the trust score to generate a new global model.

- **DNC** (Shejwalkar & Houmansadr, 2021). Singular value decomposition (SVD) is employed for Divide-and-conquer (DnC) to extract the common features. The projection of a subsampled gradients generated from a selection of a sorted set of indices is computed, and then the gradients with highest scores of outlier vector will be removed. DnC averages the gradients after repeating this process.

### A.3.4    EXPERIMENTAL RESULTS

### A.4    THE USE OF LARGE LANGUAGE MODELS

We used Large Language Models (LLMs) to aid and polish writing in this paper. Specifically, LLMs were used to help improve the clarity, grammar, and flow of certain sections of the manuscript, and to assist in refining the presentation of ideas and ensuring consistent writing style throughout the paper. All core concepts, methodological contributions, experimental designs, results, and conclusions represent our original work. The LLMs did not contribute to the research ideation, experimental methodology, data analysis, or the generation of novel scientific insights. All content assisted by LLMs was thoroughly reviewed, fact-checked, and edited by the human authors to ensure accuracy and alignment with our intended contributions. The authors take full responsibility for all claims, results, and content presented in this work.

Table 3: The comparison of the accuracy of the global model between different attacks on CIFAR10, MNIST, and Tiny ImageNet against different AGRs.

| Dataset (Model) | AGRs | No Attack(%) | Test Acc. (Difference to the Targeted Acc. ς (%)) | | | | |
|---|---|---|---|---|---|---|---|
| | | | LIE | Min-Max | Min-Sum | FMPA | ASAP |
| | | | Target Acc 55% | | | | |
| | FedAvg | 66.42 | 53.28 (-1.72) | 32.75 (-22.25) | 51.06 (-3.94) | 58.44 (3.44) | 56.37 (**1.37**) |
| | Median | 64.28 | 33.40 (-21.60) | 28.08 (-26.92) | 33.73 (-21.27) | 51.05 (-3.95) | 51.47 (**-3.53**) |
| | Trmean | 66.23 | 46.43 (-8.57) | 30.95 (-24.05) | 41.19 (-13.81) | 58.22 (3.22) | 52.50 (**-2.50**) |
| | NB | 66.73 | 51.95 (-3.05) | 45.64 (-9.36) | 55.51 (0.51) | 58.07 (3.07) | 56.42 (**1.42**) |
| | Bulyan | 66.07 | 36.91 (-18.09) | 25.95 (-29.05) | 23.52 (-31.48) | 48.71 (-6.29) | 53.71 (**-1.29**) |
| | Mkrum | 66.79 | 45.03 (-9.97) | 52.29 (-2.71) | 31.74 (-23.26) | 51.10 (-3.90) | 54.92 (**-0.08**) |
| | Fltrust | 66.59 | 31.53 (-23.47) | 50.79 (-4.21) | 52.56 (-2.44) | 53.62 (-1.38) | 55.16 (**0.16**) |
| | CC | 66.62 | 63.53 (8.53) | 10.53 (-44.47) | 14.94 (-40.06) | 58.98 (3.98) | 55.71 (**0.71**) |
| | DNC | 66.55 | 62.92 (7.92) | 63.94 (8.94) | 58.26 (3.26) | 51.43 (-3.57) | 53.14 (**-1.86**) |
| | | | Target Acc 50% | | | | |
| CIFAR10 (AlexNet) | FedAvg | 66.42 | 53.28 (6.56) | 32.75 (-34.50) | 51.06 (2.12) | 57.84 (15.68) | 50.87 (**1.74**) |
| | Median | 64.28 | 33.40 (-33.20) | 28.08 (-43.84) | 33.73 (-32.54) | 48.98 (-2.04) | 50.99 (**1.98**) |
| | Trmean | 66.23 | 46.43 (-7.14) | 30.95 (-38.10) | 41.19 (-17.62) | 52.78 (5.56) | 50.71 (**1.42**) |
| | NB | 66.73 | 51.95 (3.90) | 45.64 (-8.72) | 55.51 (11.02) | 55.79 (11.58) | 50.26 (**0.52**) |
| | Bulyan | 66.07 | 36.91 (-26.18) | 25.95 (-48.10) | 23.52 (-52.96) | 58.94 (17.88) | 49.83 (**-0.34**) |
| | Mkrum | 66.79 | 45.03 (-9.94) | 52.29 (4.58) | 31.74 (-36.52) | 62.56 (25.12) | 50.53 (**1.06**) |
| | Fltrust | 66.59 | 31.53 (-36.94) | 50.79 (1.58) | 52.56 (5.12) | 43.86 (-12.28) | 51.87 (**3.74**) |
| | CC | 66.62 | 63.53 (27.06) | 10.53 (-78.94) | 14.94 (-70.12) | 43.99 (-12.02) | 50.42 (**0.84**) |
| | DNC | 66.55 | 62.92 (25.84) | 63.94 (27.88) | 58.26 (16.52) | 51.36 (2.72) | 51.46 (**2.92**) |
| | | | Target Acc 30% | | | | |
| | FedAvg | 66.42 | 53.28 (77.60) | 32.75 (9.17) | 51.06 (70.20) | 34.49 (14.97) | 30.71 (**2.36**) |
| | Median | 64.28 | 33.40 (11.33) | 28.08 (-6.40) | 33.73 (12.43) | 27.05 (-9.83) | 29.82 (**-0.60**) |
| | Trmean | 66.23 | 46.43 (54.77) | 30.95 (3.17) | 41.19 (37.30) | 27.05 (-9.83) | 31.76 (**5.87**) |
| | NB | 66.73 | 51.95 (73.17) | 45.64 (52.13) | 55.51 (85.03) | 17.20 (-42.67) | 29.26 (**-2.47**) |
| | Bulyan | 66.07 | 36.91 (23.03) | 25.95 (-13.50) | 23.52 (-21.60) | 13.14 (-56.20) | 33.07 (**10.23**) |
| | Mkrum | 66.79 | 45.03 (50.10) | 52.29 (74.30) | 31.74 (5.80) | 27.05 (-9.83) | 31.41 (**4.70**) |
| | Fltrust | 66.59 | 31.53 (5.10) | 50.79 (69.30) | 52.56 (75.20) | 35.09 (16.97) | 33.94 (**13.13**) |
| | CC | 66.62 | 63.53 (111.77) | 10.53 (-64.90) | 14.94 (-50.20) | 36.30 (21.00) | 30.03 (**0.10**) |
| | DNC | 66.55 | 62.92 (109.73) | 63.94 (113.13) | 58.26 (94.20) | 52.09 (73.63) | 29.88 (**-0.40**) |
| | | | Target Acc 85% | | | | |
| | FedAvg | 97.98 | 94.12 (10.73) | 91.67 (7.85) | 92.84 (9.22) | 83.21 (-2.11) | 85.70 (**0.82**) |
| | Median | 97.81 | 90.99 (7.05) | 91.15 (7.24) | 92.84 (9.22) | 51.74 (-39.13) | 88.37 (**3.96**) |
| | Trmean | 97.42 | 91.80 (7.99) | 91.30 (7.41) | 92.43 (8.74) | 95.84 (12.75) | 84.45 (**-0.65**) |
| | NB | 97.96 | 88.92 (4.61) | 91.88 (8.09) | 92.29 (8.58) | 88.35 (3.94) | 86.15 (**1.35**) |
| | Bulyan | 97.97 | 92.33 (8.62) | 91.96 (8.19) | 92.29 (8.58) | 98.68 (16.09) | 87.94 (**3.46**) |
| | Mkrum | 97.94 | 95.19 (11.99) | 95.21 (12.01) | 95.39 (12.22) | 86.71 (2.01) | 85.40 (**0.47**) |
| | Fltrust | 97.96 | 87.89 (3.40) | 73.49 (-13.54) | 93.12 (9.55) | 93.00 (9.41) | 87.94 (**3.46**) |
| | CC | 97.96 | 95.35 (12.18) | 89.61 (5.42) | 94.54 (11.22) | 94.86 (11.60) | 83.72 (**-1.51**) |
| | DNC | 97.95 | 93.08 (9.51) | 92.90 (9.29) | 93.36 (9.84) | 99.47 (17.02) | 86.38 (**1.62**) |
| | | | Target Acc 80% | | | | |
| MNIST (MLP) | FedAvg | 97.98 | 94.12 (17.65) | 91.67 (14.59) | 92.84 (16.05) | 92.39 (15.49) | 80.68 (**0.85**) |
| | Median | 97.81 | 90.99 (13.74) | 91.15 (13.94) | 92.84 (16.05) | 35.52 (-55.60) | 69.70 (**-12.88**) |
| | Trmean | 97.42 | 91.80 (14.75) | 91.30 (14.13) | 92.43 (15.54) | 97.44 (21.80) | 78.75 (**-1.56**) |
| | NB | 97.96 | 88.92 (11.15) | 91.96 (14.95) | 92.29 (15.36) | 46.36 (-42.05) | 79.87 (**-0.16**) |
| | Bulyan | 97.97 | 88.92 (11.15) | 91.96 (14.95) | 92.29 (15.36) | 68.69 (-14.14) | 77.84 (**-2.70**) |
| | Mkrum | 97.94 | 95.19 (18.99) | 95.21 (19.01) | 95.39 (19.24) | 25.52 (-68.10) | 80.02 (**0.03**) |
| | Fltrust | 97.96 | 87.89 (9.86) | 73.49 (-8.14) | 93.12 (16.40) | 95.11 (18.89) | 77.47 (**-3.16**) |
| | CC | 97.96 | 94.48 (18.10) | 94.66 (18.32) | 94.54 (18.18) | 92.39 (15.49) | 76.85 (**-3.94**) |
| | DNC | 97.95 | 93.08 (16.35) | 92.90 (16.12) | 93.36 (16.70) | 92.62 (15.77) | 82.54 (**3.18**) |
| | | | Target Acc 50% | | | | |
| | FedAvg | 97.98 | 94.12 (88.24) | 91.67 (83.34) | 92.84 (85.68) | 64.02 (28.04) | 51.98 (**3.96**) |
| | Median | 97.81 | 90.99 (81.98) | 91.15 (82.30) | 92.84 (85.68) | 29.09 (-41.82) | 53.78 (**7.56**) |
| | Trmean | 97.42 | 91.80 (83.60) | 91.30 (82.60) | 92.43 (84.86) | 70.34 (40.68) | 48.06 (**-3.88**) |
| | NB | 97.96 | 88.92 (77.84) | 91.88 (83.76) | 92.29 (84.58) | 59.80 (19.60) | 50.27 (**0.54**) |
| | Bulyan | 97.97 | 92.33 (84.66) | 91.96 (83.92) | 92.29 (84.58) | 38.70 (-22.60) | 46.41 (**-7.18**) |
| | Mkrum | 97.94 | 95.19 (90.38) | 95.21 (90.42) | 95.39 (90.78) | 60.03 (20.06) | 58.33 (**16.66**) |
| | Fltrust | 97.96 | 87.89 (75.78) | 73.49 (46.98) | 93.12 (86.24) | 92.60 (85.20) | 53.08 (**6.16**) |
| | CC | 97.96 | 94.48 (88.96) | 94.66 (89.32) | 94.54 (89.08) | 92.39 (84.78) | 52.80 (**5.60**) |
| | DNC | 97.95 | 93.08 (86.16) | 92.90 (85.80) | 93.36 (86.72) | 82.40 (64.80) | 42.43 (**-15.14**) |
| | | | Target Acc 40% | | | | |
| | FedAvg | 56.46 | 51.60 (29.00) | 38.37 (-4.07) | 53.20 (33.00) | 38.79 (-3.03) | 40.42 (**1.05**) |
| | Median | 52.87 | 22.14 (-44.65) | 54.08 (35.20) | 34.24 (-14.40) | 43.46 (8.65) | 39.45 (**-1.37**) |
| | Trmean | 56.02 | 51.60 (29.00) | 54.59 (36.47) | 39.82 (-0.45) | 46.95 (17.38) | 40.12 (**0.30**) |
| | NB | 57.23 | 52.98 (32.45) | 52.95 (32.37) | 53.09 (32.73) | 34.09 (-14.78) | 38.74 (**-3.15**) |
| | Bulyan | 56.03 | 24.93 (-37.67) | 48.01 (20.02) | 33.51 (-16.23) | 38.90 (-2.75) | 39.52 (**-1.20**) |
| | Mkrum | 54.98 | 27.02 (-32.45) | 49.68 (24.20) | 26.39 (-34.03) | 35.61 (-10.98) | 40.82 (**2.05**) |
| | Fltrust | 55.63 | 33.57 (-16.07) | 47.04 (17.60) | 53.45 (33.63) | 35.97 (-10.08) | 37.86 (**-5.35**) |
| | CC | 53.23 | 29.13 (-27.17) | 32.26 (-19.35) | 30.99 (-22.53) | 41.76 (4.40) | 40.91 (**2.27**) |
| Tiny ImageNet (ResNet50) | DNC | 53.13 | 68.12 (70.30) | 69.66 (74.15) | 54.29 (35.73) | 40.64 (1.60) | 39.31 (**-1.73**) |
| | | | Target Acc 35% | | | | |
| | FedAvg | 56.46 | 51.60 (47.43) | 38.37 (9.63) | 53.20 (52.00) | 48.06 (37.31) | 33.73 (**-3.63**) |
| | Median | 52.87 | 22.14 (-36.74) | 54.08 (54.51) | 34.24 (-2.17) | 33.88 (-3.20) | 34.45 (**-1.57**) |
| | Trmean | 56.02 | 51.60 (47.43) | 54.59 (55.97) | 39.82 (13.77) | 48.47 (38.49) | 34.35 (**-1.86**) |
| | NB | 57.23 | 52.98 (51.37) | 52.95 (51.29) | 53.09 (51.69) | 50.12 (43.20) | 32.73 (**-6.49**) |
| | Bulyan | 56.03 | 24.93 (-28.77) | 48.01 (37.17) | 33.51 (-4.26) | 4.27 (-87.80) | 37.75 (**7.86**) |
| | Mkrum | 54.98 | 27.02 (-22.80) | 49.68 (41.94) | 26.39 (-24.60) | 37.30 (6.57) | 34.51 (**-1.40**) |
| | Fltrust | 55.63 | 33.57 (-4.09) | 47.04 (34.40) | 53.45 (52.71) | 49.27 (40.77) | 35.34 (**0.97**) |
| | CC | 53.23 | 29.13 (-16.77) | 32.26 (-7.83) | 30.99 (-11.46) | 41.70 (19.14) | 37.16 (**6.17**) |
| | DNC | 53.13 | 68.12 (94.63) | 69.66 (99.03) | 54.29 (55.11) | 48.34 (38.11) | 35.49 (**1.40**) |

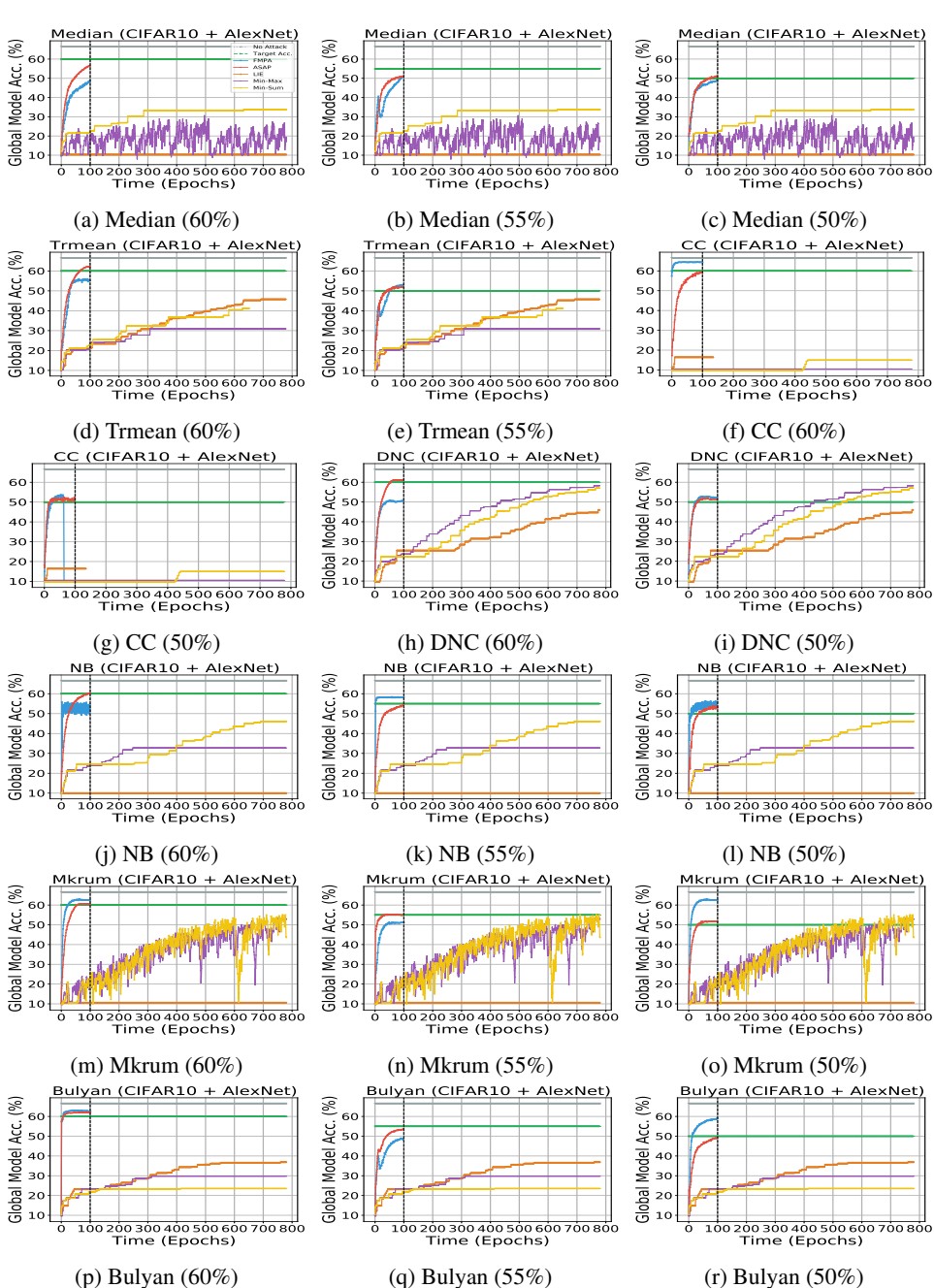

Figure 5: Comparison of each attack against various AGRs with different attack objectives on CIFAR10 with AlexNet under different attack objectives and different attacks under the same target accuracy.

