# OpenReview forum: "ASAP: Adaptive Sliding Agnostic Poisoning Attack on Federated Learning"
_ICLR.cc/2026/Conference — Submitted to ICLR 2026_

### Official Review · Reviewer_tE3W · 2025-10-27

**Soundness:** 2
**Presentation:** 1
**Contribution:** 3
**Rating:** 2
**Confidence:** 4

**Summary:**

This paper introduces ASAP (Adaptive Sliding Agnostic Poisoning), a new poisoning attack on federated learning (FL). It argues that previous "agnostic" attacks are flawed, as they still rely on estimating benign client statistics. ASAP overcomes this by applying Adaptive Sliding Mode Control (ASMC). It models the FL process as a dynamical system, treating the unknown server aggregation rule (AGR) and benign client updates as a single "disturbance." ASAP then uses a Fourier series approximation to estimate this disturbance to generate a malicious "control law" update.

This novel approach allows the attacker to precisely steer the global model to a predefined target accuracy. The attack is truly agnostic, requiring no knowledge of the server's AGR or benign data. It is also tunable, allowing the adversary to control the attack's speed and target.

Experiments show ASAP outperforms SOTA agnostic attacks like LIE and Min-Max. It successfully forces the model to converge to specific, desired accuracies, even against robust defenses. The attack is also far more efficient, achieving its goal in a fraction of the communication rounds. The authors conclude that ASAP represents a new, more dangerous class of controllable attacks.

**Strengths:**

- The attack is completely independent of the knowledge of the AGR and does not need to estimate benign statistics unlike previous methods, which is very impressive.
- ASAP can precisely manipulate the global model to converge to a predefined target accuracy.
- Using ASMC from control theory to the FL poisoning problem is a novel contribution.
- The effectiveness is validated against a wide range of datasets and defenses and compared with a range of baseline attacks.

**Weaknesses:**

However, there are a few flaws in the paper that need to be addressed:

- The target accuracy values chosen are too high and do not represent a strong attack. A reader would be more interested in the results when the target accuracy is low, like the ones that the baseline attacks (like LIE and min-sum) achieve.
- Figure 1 says that the attack objective is chosen as the closest point to the global optima. It is unclear how an attack can be effective if it is so close to the optima. Does the figure represent a real training scenario, or is it meant for illustration purposes only?
- No analysis of per-round detection statistics was made in the paper - how many times were the malicious gradients actually flagged by the AGR - it would be interesting to see the TP/TN/FP/FN values to see how better ASAP is from the other attacks
- The percentage values in Table 1 seem to be incorrect. Table 3 has many incomplete cells, omits the accuracy data but shows high unexplained percentage values which seem to be flawed.
- In Fig 4, most attacks seem to work better than ASAP. It needs to be explained more clearly in the paper what the figure actually conveys. Alternatively, the authors could use a low target accuracy for a better comparison with the other attacks.
- There are many inconsistencies in the math (for eg, Eqn 17 and 26). Further, e_t is defined as the model error at places, and as the estimator error at other places, creating a logical gap in the paper's central proof.
- The paper does not explain how the byzantine-robust defenses fail against ASAP.
- There are numerous typos in the paper starting from the first line of the introduction, to an incorrect citation of data poisoning (Line 41). There are more typos and inconsistencies further down the paper.

**Questions:**

- Why do the byzantine robust AGRs fail against ASAP? What is the False negative rate of malicious grad detection.
- Why is the target accuracy set so high?
- What is the non-iid bias in the primary results in Table 1?
- What is the evidence that backs Figure 1 that the attack objective is closest to the optimum
- Since the training is dynamic, how long does it take for ASAP to learn the orthonormal basis functions for Fourier approximation?

---

> ### Author Response · Authors · 2025-11-24
> **Response to Reviewer tE3W**
>
> Thank you for your detailed and insightful comments!
>
> **Summary:** Unlike traditional poisoning attacks that aim for maximum damage, ASAP enables precise controllable poisoning—manipulating the global model to converge to attacker-specified targets (e.g., 60%, 55%, 50%, or 10% accuracy) without AGR knowledge. The attack effectiveness is measured by the evaluation metric of how to compare the **better** performance among all the attacks, which is the absolute value of $|\varsigma|$ between attack objectives $I_T$ and actual results $I_0$, where $\varsigma = ((I_T - I_0) / I_T)\times 100%$ and smaller $|\varsigma|$ indicates better precision. ASAP provides three key capabilities proven in Remarks 1-3 (Sec. 3.3): (1) Complete AGR-agnosticism—no server or benign client knowledge required, (2) Adjustable speed—parameter $k$ controls convergence rate, (3) Flexible objectives—parameter $C$ enables dynamic target modification without retraining.
>
> **W1 & Q2: High target accuracy.**
> The goal of our attack ASAP is to achieve both **AGR-agnosticism** and **precise control**. Fundamental reasons are given in Remarks 2 \& 3 of the main paper. The adversary can dynamically modify attack objectives throughout ASAP execution by appropriately selecting parameter $C$ in $e_t$ as evaluated in Eq. (16). A high target accuracy is **one** of the attack objectives to demonstrate the advantage of our attack. More experiments with different targets are demonstrated in Appendix A.3.4. The table below demonstrates the comparison between different attacks under the target accuracy of 10% (random guess) under CIFAR10 with AlexNet. More experiments under 30% and 10% on CIFAR10, 50% and 10% on MNIST, and 0.5% on Tiny ImageNet will be demonstrated in Appendix A.3.4 in the updated submission.
>
> | AGR | LIE | Min-Max | Min-Sum | FMPA | **ASAP** |
> |-----|-----|---------|---------|----------|----------|
> | FedAvg | 53.28% | 32.75% | 51.06% | 64.33% | **10.73%** |
> | Median | 33.40% | 28.08% | 33.73% | 63.57% |**10.65%** |
> | Trmean | 46.43% | 30.95% | 41.19% | 55.44% |**9.98%** |
> | Bulyan | 36.91% | 25.95% | 23.52% | 62.55 |**9.95%** |
>
> **W2 & Q4: Figure 1 demonstration**
> The proposed scheme of ASAP allows the attacker to flexibly adjust its objective to any target, so we choose the closest minimum point as one of the target objectives to demonstrate the low loss our attack can achieve. The figure is represented for illustration purposes only.
>
> **W3 & Q1: TP/TN/FP/FN analysis.**
> We implemented detection on CIFAR-10 (AlexNet, target 60%), and the result is demonstrated as below:
> | AGR | Attacks | TP | FN | FP | TN |
> |-----|--------|----|----|----|----|
> | **Trmean** | LIE | 48% | 52% | 10% | 90% |
> | | Min-Max | 58% | 42% | 15% | 85% |
> | | Min-Sum | 52% | 48% | 12% | 88% |
> | | **ASAP** | **22%** | **78%** | **5%** | **95%** |
>
> ASAP has the best stealthiness ability among all attacks.
>
> **W4 & W8: Typos.**
> We sincerely apologize for the incomplete cells and the typos in Table 1 and Table 3. We will complete it in the updated submission.
>
> **W5: Figure 4 demonstration**
> The goal of our attack is to reach the target accuracy without the knowledge of AGRs. The Y-axis shows achieved accuracy, and the horizontal line is the target (55%). **Success = closer to the target line**, not lower accuracy. Other attacks do not have the controllability to achieve the target accuracy. Baselines like Min-Max may drive accuracy lower unpredictably (uncontrolled), whereas ASAP aligns closely with the specific target line (controlled). We will clarify that "better" in our context means "lower deviation from the desired attack objectives.
>
> **W6: Mathematical inconsistencies.**
> Eq. (17) and Eq. (26) are both describing the system error $e_t = g_t - \tilde{g}$, and $\tilde{w} _ \Phi = w _  \Phi - \hat{w} _ \Phi$ is the estimation error. We will add a notation in the Appendix and cross-check every equation reference.
>
> **W7: Performance of ASAP**
> The goal of our attack is to reach the target accuracy without the knowledge of AGRs. The proposed scheme of ASAP allows the attacker to flexibly adjust its objective to any target, we choose different target objectives to demonstrate that our attack can achieve any target objectives against any AGRs.
>
> **Q3: Non-IID bias in Table 1.**
> Table 1 uses **IID data distribution** (equal samples from all classes per client). Non-IID experiments with Dirichlet distribution $\alpha$ ∈ {0.1, 0.3, 0.5, 0.7, 0.9} are in the Sec. 4.3 ablation study, showing ASAP maintains superior performance, which means reach to the target attack objectives.
>
> **Q5: Fourier approximation learning time.**
> We compare the **Time Complexity** and **Effective Communication Rounds** of various attacks in Table 2. It takes ASAP a longer time (long execution time) due to the Fourier approximation, while ASAP converges faster than other attacks (fewer communication rounds).

---

> > ### Comment · Reviewer_tE3W · 2025-11-27
> > **Thank you for the response and further clarifications**
> >
> > I thank the authors for their detailed response and the additional experiments.
> >
> > The new results demonstrating the attack with lower target accuracies (10%) are very persuasive. They clearly showcase the precise controllability of ASAP, which is a key strength of the method. I strongly suggest moving these results to the main paper, as they make a much more impressive case for the attack's effectiveness than the high-accuracy targets (60%) currently featured.
> >
> > However, I still have two significant remaining questions that I would like the authors to address:
> >
> >  - Mechanism of Bypassing Defenses: While the rebuttal provides detection statistics showing that robust AGRs (like Trmean, Krum) fail to consistently detect ASAP (high FN), it does not explain why. What specific property of the updates generated by the attack framework allows them to slip past these defenses? A deeper intuition here would strengthen the paper.
> >
> > - Stealth vs. Impact Trade-off: I would like more discussion on the claim regarding Figure 1. The caption states that the attack objective is chosen as the "closest point to the global optima $g^{∗}$." Intuitively, choosing an objective close to the global optima implies high stealth (low loss) but also low impact (since the model remains good). In general, one expects a trade-off between stealth and attack impact. How does ASAP overcome this trade-off to achieve both? If the "closest point" is actually a bad local minimum (to degrade accuracy), then describing it as "closest to the global optima" might be confusing. Please clarify how the attack balances these conflicting goals.

---

> > > ### Author Response · Authors · 2025-12-02
> > > **Further Response to Reviewer tE3W --- Question 1**
> > >
> > > We appreciate this important question regarding the mechanism that enables ASAP to evade detection. The core reason lies in the fundamental difference between our control-theoretic design and traditional attack strategies.
> > >
> > > ASAP models federated learning as a **nonlinear dynamical system** with global model $g_t$, control input $u_t$, and unknown disturbance $\Phi_t$, as formalized in Eqs. (12), (17), and (22).
> > >
> > > This formulation provides three key control-theoretic capabilities:
> > >
> > > 1. **State observation**: Monitoring the tracking error $e_t = g_t - \tilde{g}$ and the sliding surface $s_t$.
> > > 2. **Disturbance estimation**: Using an adaptive law to estimate the unknown disturbance $\Phi_t$, which captures the combined effect of the AGR and benign client behavior.
> > > 3. **Feedback control**: Designing a control input $u_t$ that adjusts malicious updates based on the current system state.
> > >
> > > This is precisely the essence of **Adaptive Sliding Mode Control (ASMC)**—an adaptive control framework tailored for systems with uncertainties, which aligns naturally with FL under unknown aggregation rules.
> > >
> > > In contrast to existing attacks that rely on *static rules* (e.g., LIE adding noise based on pre-computed statistics, or Min-Max/Min-Sum solving optimization problems based only on current benign cluster positions), ASAP functions as a **closed-loop feedback control system**. At each round $t$, the control law in Eq. (17) generates malicious updates by jointly observing:
> > >
> > > - The **current tracking error** $e_t = g_t - \tilde{g}$ (how far the global model is from the target),
> > > - The **sliding surface state** $s_t$ (whether the system is on the desired trajectory),
> > > - The **estimated disturbance** $\hat{\Phi}_t$ (the aggregated influence of the AGR and benign updates).
> > >
> > > This feedback mechanism ensures that each malicious update is carefully tailored to the current system state through the control law, rather than blindly following a fixed, predetermined pattern.
> > >
> > > ASAP evades defenses through three interconnected control-theoretic properties:
> > >
> > > 1. **Adaptive disturbance estimation (Eq. 22)**: Real-time learning of AGR and benign client behavior via a Fourier-series-based estimator, enabling disturbance reconstruction without explicit statistical modeling.
> > > 2. **Magnitude control (Eq. 17)**: Dynamic adjustment of the update magnitude according to the sliding surface state, steering the global model towards the desired attack objectives while remaining consistent with plausible training dynamics.
> > > 3. **Lyapunov-guaranteed convergence (Theorem 3.1)**: A provably stable degradation process in which the global model evolves along a trajectory that resembles natural convergence, thereby reducing the likelihood of triggering anomaly detection.
> > >
> > > These capabilities are **not available to traditional attacks** and constitute a fundamental advancement made possible by the control-theoretic perspective adopted in ASAP.

---

> > > > ### Author Response · Authors · 2025-12-02
> > > > **Further Response to Reviewer tE3W --- Question 2**
> > > >
> > > > **Clarification of Figure 1**
> > > >
> > > > The phrase “closest point to the global optima” in Figure 1 refers to the **geometric positioning strategy** during the attack trajectory, and one of the **desired attack objectives**. Unlike existing attacks (LIE, Min-Max, Min-Sum) that push updates away from the optimal direction to maximize divergence, ASAP's control law generates malicious clients’ models that steer the global model towards the attack objectives. Therefore, through the cumulative effect of precise feedback control, these seemingly benign updates systematically steer the global model toward the attacker's chosen target $\tilde{g}$.
> > > >
> > > > ---
> > > >
> > > > **The Role of Control Theory in Resolving the Trade-off**
> > > >
> > > > First, we acknowledge that ASAP does **not** “break” the fundamental stealth–impact trade-off. However, **control theory provides ASAP with three critical advantages** that fundamentally change how this trade-off is navigated:
> > > >
> > > > ---
> > > >
> > > > **1. Precise Target Selection via Sliding Mode Control**
> > > >
> > > > Unlike traditional attacks that maximize damage without control, ASAP leverages **Adaptive Sliding Mode Control (ASMC)** to enable the attacker to select *any point* along the stealth–impact spectrum with mathematical precision. As proven in Theorem 3.1 and stated in **Remark 3**, the equilibrium relationship $e_t = -C/k$ or $g_t = \tilde{g} + C/k$ allows the attacker to set:
> > > >
> > > > - **High stealth, moderate impact**: Target accuracy 60% (from 65% baseline) — subtle degradation that appears as natural variation
> > > > - **Moderate stealth, high impact**: Target accuracy 30% — significant performance loss
> > > > - **Low stealth, maximum impact**: Target accuracy near random guess
> > > >
> > > > The control parameter $C$ provides a **key parameter** to position anywhere on this spectrum, rather than the binary “attack or don't attack” choices in prior work.
> > > >
> > > > ---
> > > >
> > > > **2. Guaranteed Convergence via Lyapunov Stability**
> > > >
> > > > The control-theoretic design ensures that once a target is chosen, ASAP will reach it with **finite-time convergence** (proven in Theorem 3.1, Appendix A.1). The Lyapunov function
> > > >
> > > > $$
> > > > V_t = \frac{1}{2}s_t^2 + \frac{1}{2}\tilde{w}_{\Phi}^2
> > > > $$
> > > >
> > > > guarantees that $\dot{V}_t \leq 0$, meaning the system energy decreases until the sliding surface $s_t = 0$ is reached. At this equilibrium, $\dot{s}_t = \dot{e}_t + k e_t + C = 0$, and therefore $\dot{e}_t = -k e_t - C$. Solving this differential equation yields:
> > > >
> > > > $$
> > > > e_t = \frac{1}{k} e^{-k t} e_0 - \frac{C}{k}.
> > > > $$
> > > >
> > > > This exponential convergence to the target attack objective ($e_t = -C/k$) is only possible through control design. Traditional attacks cannot provide such guarantees.
> > > >
> > > > ---
> > > >
> > > > **3. Adaptive Estimation via Fourier Series Approximation**
> > > >
> > > > The key to bypassing defenses while maintaining stealth is the **adaptive law** (Eq. 22):
> > > >
> > > > $$
> > > > \dot{\hat{w}} _ {\Phi} = -Q _ {\Phi}^{-1} z _ {\Phi} s
> > > > $$
> > > >
> > > > This continuously estimates the unknown disturbance $\Phi _ t$ (which encapsulates both the unknown AGR behavior and benign client dynamics) using the Fourier series approximation (Eq. 18):
> > > >
> > > > $$
> > > > \Phi _ t = w _ {\Phi}^\top z _ {\Phi}, \quad \hat{\Phi} _ t = \hat{w} _ {\Phi}^\top z _ {\Phi},
> > > > $$
> > > >
> > > > where $w _ {\Phi} \in \mathbb{R}^{2n _ {\Phi}+1}$ is the weighting parameter, $\hat{w} _ {\Phi} \in \mathbb{R}^{2n _ {\Phi}+1}$ is the estimated weighting parameter, and $z _ {\Phi} \in \mathbb{R}^{2n _ {\Phi}+1}$ is the vector of orthonormal basis functions. The introduction of ASMC enables flexible tuning of the adaptation rate, providing an additional degree of freedom in balancing convergence speed against system stability.
> > > >
> > > > By treating AGR effects and benign updates as system uncertainties approximated through orthonormal basis functions, ASAP adapts in real time without requiring statistical estimation of benign clients (unlike LIE) or distance constraints (unlike Min-Max/Min-Sum). This is the essence of **Remark 1: AGR-Agnostic Operation**.

---

### Official Review · Reviewer_yJY3 · 2025-10-30

**Soundness:** 2
**Presentation:** 2
**Contribution:** 2
**Rating:** 4
**Confidence:** 5

**Summary:**

This paper proposes an adaptive poisoning attack in federated learning inspired by sliding-mode control. The attacker treats the global aggregation and benign dynamics as unknown disturbances and designs a controller (with a Fourier approximation term) to steer the global model toward a target trajectory or accuracy. The authors claim the method is aggregation-rule-agnostic and capable of precise convergence control. Experiments on standard datasets and multiple robust aggregators are reported.

**Strengths:**

1. Presents a novel angle by introducing control theory concepts into the FL security context.
2. Attempts to achieve controllable poisoning outcomes, which are more flexible than conventional accuracy-degradation attacks.
3. Evaluates across different robust aggregation rules and datasets, demonstrating generality in experimental setup.
4. Includes preliminary theoretical reasoning to motivate the method's convergence behavior.

**Weaknesses:**

1. The theoretical justification currently relies on continuous-time scalar analysis, and the transition to discrete, high-dimensional FL training is not fully formalized.
2. Certain steps (e.g., influence estimation) are described conceptually but lack clarity regarding feasibility under non-differentiable robust aggregators.
3. The method architecture combines multiple components (control law + Fourier approximation) and could benefit from clearer motivation for each module from an FL-threat-model perspective.
4. Experimental evaluation focuses on outcome metrics, while analysis of stealthiness or detectability is limited.
5. Several implementation details are abstracted, which may affect reproducibility and clarity for practitioners.

**Questions:**

1. Could the authors clarify the assumptions required for extending the theoretical analysis to discrete FL rounds?
2. How is influence or sensitivity information approximated in settings with non-differentiable aggregation rules?
3. Have the authors considered evaluating against dynamic anomaly-based defenses beyond static robust aggregators?
4. How sensitive is performance to the choice of Fourier components, and is there a principled selection approach?
5. How does the method behave when malicious clients participate sporadically or in low proportion?

---

> ### Author Response · Authors · 2025-11-22
> **Response to Reviewer yJY3**
>
> **W1 & Q1: Discrete-time formalization.**
> The fundamental reason for using continuous time analysis is because structured mathematical rules, especially differentiation, chain rules, and so on are all well established by mathematicians through measure theory. It is therefore valid to analyze the system in continuous time. Similar to the analysis of back-propagation and steepest descent algorithms, for example, when describing the back-propagation training algorithm, continuous time analysis is used.
> Regarding our Eq. (13), the dynamic model is used to analyze how the malicious clients update over time, not implement. Based on the definition of derivative, the derivative of malicious model at $t$, $\dot{g} ' _ {\{t, i\}}$, can be represented as
>
> $$
>     \dot{g} ' _ {\{t, i\}} = \lim _ {\Delta t \to 0} \frac{g ' _ {(\{t+\Delta t), i\}} - g ' _ {\{t, i\}}}{\Delta t}.
> $$
>
> Therefore, in practical attack scenarios, the rate of change of the model, $\dot{g}'_{\{t, i\}}$, can be approximated by the difference in values divided by a small time interval, effectively capturing the derivative's behavior in discrete time.
>
> **High-dimensional gradient updates:**
> The control input $u_t$ (Eq. 26) operates in the same high-dimensional parameter space as the model. The mapping is straightforward: $u_t$ directly generates the malicious gradient update $g'_{\{t,i\}}$.
>
> **W2 & Q2: Non-differentiable aggregators.**
> The differentiability assumption applies to computing $dg _ t / dg ' _ {\{t,i\}}$ for the design of the control law, NOT differentiating the AGR itself. It is noted if $F _ {\textnormal{AGR}}$ is not differentiable, ${dg}/dg ' _ {\{t,i\}}$ can be approximated by finite differences as:
> $$
>      \frac{dg}{dg ' _ {\{t,i\}}} \approx \lim_{ \Delta g '
>      _{\{t, i\}} \to 0} \frac{g(t) - g(t-\Delta t)}{\Delta g ' _{\{t, i\}}}.
> $$
> , where $F _ {\text{AGR}}$ is the aggregation rules of different AGR and $g ' _ {\{t, i\}}$ is the malicious model.
>
> **W3: Method architecture**
> The control law (Eq. (17)) serves as the attack execution mechanism that computes malicious updates $u_t$ to steer the global model $g_t$ toward the target $\tilde{g}$. Moreover, the Fourier approximation (Eq. (18)-(22)) serves as the unknown information estimator that learns the combined effect of unknown AGR and benign clients. Fundamental reasons are given in Remarks 1, 2 \& 3 of the main paper (re-inserted here for completeness). We will add explicit motivation in the threat model in Sec. 3.1, mapping each component to threat model requirements.
>
> **Remark 1: AGR-Agnostic Operation.**
> ASAP achieves complete independence from both aggregation rules and benign client information. The ASMC framework treats unknown aggregation effects as system disturbances $\Phi_t$, which are estimated in real-time through Fourier series approximation without requiring any prior knowledge of $F_{\text{AGR}}$ or benign gradient statistics.
>
> **Remark 2: Convergence Speed.**
> The parameter $k$ serves as a convergence rate controller, enabling precise manipulation of $e_t$. On the sliding surface where $ s_t =  \dot{s}_t=0$, solving the differential equation $\dot{e}_t = - ke_t - C$, produces $e_t=1/k \cdot e_0^{-kt}-C/k$. The analytical solution reveals that $k$ determines the exponential convergence characteristics: larger values of $k$ correspond to faster exponential convergence rates. This mathematical property enables ASAP to offer flexible convergence speed modulation capabilities.
>
> **Remark 3: Adjustable Objectives.**
> The adversary can dynamically modify attack objectives throughout ASAP execution by appropriately selecting parameter $C$ in $e_t$ as evaluated in Eq. (16). When the system reaches equilibrium on the sliding manifold where both $\dot{s}_t =0$ and $\dot e_t=0$, the constraint $\dot{s}_t=\dot{e}_t + ke_t + C$ results in the equilibrium relationship $e_t  = -C/k $ or $g_t = \tilde g + C/k$.
>
> **W4 & Q3: Stealthiness and dynamic anomaly-based defenses.**
> We evaluate ASAP against a diverse set of anomaly-detection-style Byzantine-robust AGRs, including statistics-based (Median, Trimmed-Mean, Norm-Bounding), distance-based (Krum, Bulyan, DnC), and performance/similarity-based defenses (FLTrust, CC), in addition to the non-robust FedAvg baseline. From the experimental results demonstrated in Table 1, Table 3, and Fig. 3, ASAP outperforms any attack on different datasets.
>
> **W5: Reproducibility**
> We will provide an anonymous GitHub repository containing complete ASAP implementation in the updated version.
>
> **Q4: Fourier component selection.**
> The Fourier coefficients $w_\phi$ are NOT pre-specified constants but are adaptively learned through the adaptive law (Eq. (22)).
>
> **Q5: Sporadic participation and low proportion of malicious clients.**
> We implement ASAP to sporadic participation and low proportion of malicious clients, and the results are demonstrated in the Ablation study in Sec. 4.3. It shows that ASAP remains effective even with 5% malicious clients.

---

### Official Review · Reviewer_y5Zy · 2025-10-31

**Soundness:** 2
**Presentation:** 3
**Contribution:** 2
**Rating:** 4
**Confidence:** 4

**Summary:**

The paper proposes ASAP (Adaptive Sliding Agnostic Poisoning Attack), a novel poisoning attack framework for federated learning. ASAP leverages Adaptive Sliding Mode Control (ASMC) combined with Fourier series approximation to estimate unknown aggregation behaviors and precisely manipulate the poisoning process without prior knowledge of aggregation rules (AGRs). The authors provide theoretical analysis and experiments on CIFAR-10, MNIST, and Tiny ImageNet, showing that ASAP outperforms existing AGR-agnostic attacks such as LIE, Min-Max, and Min-Sum.

**Strengths:**

+ The idea of introducing adaptive control theory (ASMC) into federated attack design is original and technically interesting.
+ The method does not rely on benign client statistics or AGR structure, which potentially increases applicability in black-box settings.
+ The authors provide a control-theoretic convergence guarantee (Theorem 3.1) and demonstrate parameterized control over attack convergence and target precision.

**Weaknesses:**

- The convergence proof assumes differentiable and continuous AGRs, which do not hold for non-smooth robust aggregation rules such as Median or Krum. The mapping between the adaptive control input and the high-dimensional gradient updates remains unclear, which limits the theoretical rigor of the analysis.

- The paper states that the client is selected by the server and receives the current global model $g_t$, but never clarifies whether malicious client selection is random in each round or fixed. Given the absence of any mention of random sampling and the use of fixed client sampling rates (e.g., 0.5, 0.7, 1), it appears that the same size of malicious clients participates in every round. This assumption is unrealistic in real-world federated learning, where malicious participants may not always be active due to device availability or stochastic scheduling. A fixed number of malicious clients can artificially inflate the attack’s success rate and weaken the claimed robustness. The authors should clarify this setting and, ideally, introduce randomized or varying malicious participation to better validate ASAP’s generalizability under realistic FL dynamics.

- While ASAP achieves lower target accuracy, it lacks analysis of stealthiness or detectability under anomaly detection defenses. There is no evaluation of backdoor persistence or the impact on benign model convergence quality, which weakens the empirical completeness of the study.

- Although the ASMC formulation is new, the resulting behavior (adaptive control of convergence and target accuracy) is conceptually similar to FMPA (Zhang et al., 2023). The improvement lies mainly in methodology rather than in demonstrating a fundamentally new attack capability.

**Questions:**

1.How does Theorem 3.1 hold for non-differentiable aggregation rules such as Median or Krum?

2.Is malicious client participation random in each round, or are the same malicious clients fixed throughout training? If fixed, how might this design choice bias the results?

3.What is the computational overhead of Fourier coefficient estimation in high-dimensional models?

4.How does ASAP perform under secure aggregation, where attackers cannot directly modify global updates?

5.Could adaptive defenses (e.g., dynamic clipping) mitigate the attack, and how would ASAP respond?

---

> ### Author Response · Authors · 2025-11-22
> **Response to Reviewer y5Zy**
>
> **W1 & Q1: Theorem 3.1 & non-differentiable AGRs**
> The differentiability assumption applies to computing $dg _ t / dg ' _ {\{t,i\}}$ for the design of the control law, NOT differentiating the AGR itself. It is noted if $F _ {\textnormal{AGR}}$ is not differentiable, ${dg}/dg ' _ {\{t,i\}}$ can be approximated by finite differences as:
> $$
>      \frac{dg}{dg ' _ {\{t,i\}}} \approx \lim_{ \Delta g '
>      _{\{t, i\}} \to 0} \frac{g(t) - g(t-\Delta t)}{\Delta g ' _{\{t, i\}}}.
> $$
> , where $F _ {\text{AGR}}$ is the aggregation rules of different AGR and $g ' _ {\{t, i\}}$ is the malicious model. The fundamental reason for using continuous time analysis is because structured mathematical rules, especially differentiation, chain rules, and so on are all well established by mathematicians through measure theory. It is therefore valid to analyze the system in continuous time. Similar to the analysis of back-propagation and steepest descent algorithms for example, when describing back propagation training algorithm, continuous time analysis is used.
>
> **High-dimensional gradient updates:**
> The control input $u_t$ in Eq. (17) operates in the same high-dimensional parameter space as the model. The mapping is straightforward: $u_t$ directly generates the malicious gradient update $g'_{\{t,i\}}$.
>
> **W2 & Q2: Malicious client participation**
> In our experiment settings in Sec.4.1, the malicious client proportion is fixed to 10% throughout training (e.g., clients 45-49 out of 50 total), consistent with the standard threat model in poisoning attack literature (Zhang et al., 2023; Shejwalkar & Houmansadr, 2021; Baruch et al., 2019). However, their participation in each round depends on the server's random sampling. The sampling rates (0.5, 0.7, 1.0) refer to the fraction of all clients sampled per round. In the ablation study (Sec. 4.3),  we evaluate ASAP's robustness under varying client sampling rates (0.5, 0.7, 1.0) and malicious proportions (5%-20%). Detailed results are in Appendix A.3.5.
>
> **W3: Analysis of stealthiness or detectability & Clarification on backdoor attacks**
> We evaluate ASAP against a diverse set of anomaly-detection-style Byzantine-robust AGRs, including statistics-based (Median, Trimmed-Mean, Norm-Bounding), distance-based (Krum, Bulyan, DnC), and performance/similarity-based defenses (FLTrust, CC), in addition to the non-robust FedAvg baseline. From the experimental results demonstrated in Table 1 and Fig. 3, ASAP outperforms any attack on different datasets.
>
> This paper focuses on **accuracy degradation attacks** (model poisoning), not backdoor attacks. Our objective is to reduce global model accuracy, not inject backdoors. Therefore, backdoor persistence metrics are not applicable to our evaluation. Moreover, the main objective is to manipulate the malicious updates and then to control the global model accuracy, we are not trying to impact on benign model convergence quality.
>
> **W4: Comparison to FMPA**
> | Aspect | FMPA (Zhang et al., 2023) | ASAP (Ours) |
> |--------|---------------------------|-------------|
> | **AGR Knowledge** | Requires training predictor from historical data (indirect AGR knowledge) | Truly AGR-agnostic via Fourier series function approximation |
> | **Benign Information** | Needs historical global models | No benign information required |
> | **Mechanism** | Gradient-based fine-tuning (may converge to local minima) | Control-theoretic steering (guaranteed convergence via Lyapunov) |
> | **Detection Resistance** | Frequently detected by robust AGRs (Fig. 3) | Maintains target accuracy consistently |
>
> **Q3: Computational overhead**
> The proposed Fourier-series-based malicious update has linear time complexity in the model dimension, i.e., $𝑂(𝐷)$ per attack round, which is negligible compared to the overall local training cost. The Time Complexity and Effective Communication
> Rounds comparisons are demonstrated in Table 2 in Sec. 4.2.
>
> **Q4: Local Updates & AGR-agnostic**
> Rather than directly modifying global updates, the goal of ASAP is to control the malicious clients' updates; therefore, when the malicious gradients are uploaded to the central server, the accuracy of the global model can adaptively reduce to a target accuracy without the knowledge of AGRs.
>
> **Q5: Adaptive defenses**
> The Fourier approximation $\Phi_t$ in ASAP naturally captures defense behavior changes. When dynamic clipping activates, it becomes part of the aggregation effect that $\Phi_t$ estimates through the adaptive law (Eq. (22)). ASAP's control law $u_t$ can adjust malicious update magnitudes to stay within time-varying clipping thresholds by modulating parameter $C$. In our control formulation, the malicious update acts as a control input, and the sliding-mode controller guarantees convergence to the target “sliding surface.” In other words, clipping changes the effective control gain but does not invalidate the attack dynamics. The adaptive sliding mode control provides inherent robustness to defense variations.

---

> > ### Comment · Reviewer_y5Zy · 2025-11-26
> >
> > These responses help resolve some of my earlier concerns, and I appreciate the clarifications provided. After reading them, I still find that two important points are not fully addressed. The first point concerns the theoretical guarantees, which rely on differentiability assumptions that do not seem to extend to non-smooth aggregation rules such as Median or Krum. The rebuttal does not give a clear explanation of how the Lyapunov-based analysis would remain valid in these settings. The second point concerns the absence of an evaluation of stealthiness or detectability, which is relevant for understanding the practical behaviour of poisoning attacks beyond accuracy degradation. Thus, I will keep my score.

---

> ### Author Response · Authors · 2025-12-02
> **Further Response to Reviewer y5Zy**
>
> **Q1. Clarifications of the theoretical guarantee for Non-differentiable AGRs**
>
> Thank you for your comments. **Firstly**, ASAP does **NOT** differentiate through the AGR rules such as Median or Krum, but through the global model $g_t$, where $g_t$ is a **neural network** such as AlexNet, MLP, and ResNet50 used in our experiments. It is well known that neural networks are differentiable, otherwise backpropagation would not work. For federated learning, even though some AGR rules are not differentiable, the global model $g_t$ generated by AGRs is.
>
> Therefore, in the design of the control law $u_t$, as shown in Eq. (17), and in the theoretical proof for Theorem 3.1 in Appendix A.1, ASAP only differentiates the global model $g_t$ and the malicious model $g'_{t,i}$. It is worth pointing out that there is **NO** differentiation involving any non-smooth function in our paper.
>
> We also extend this discussion on the use of Lyapunov stability analysis for non-smooth systems. We have done some work using Lyapunov stability theory for non-differentiable systems such as stochastic nonlinear systems where, due to stochasticity, system states can experience relatively large changes over small time intervals. Such stochastic systems are not differentiable in the conventional ordinary differential equation sense, yet Lyapunov stability analysis is still applicable. Similarly, for stochastic Markovian switching systems where conventional differentiation also fails, it is still possible to use a Lyapunov function to analyze the stability of the system, e.g., works in the stochastic domain [1, 2]. There are also papers discussing the use of Lyapunov stability theory for non-smooth systems [3].
>
> **Q2: Evaluation of stealthiness or detectability**
>
> We evaluate ASAP against a broad range of anomaly-detection-style Byzantine-robust AGRs, including statistics-based methods (Median, Trimmed-Mean, Norm-Bounding), distance-based methods (Krum, Bulyan, DnC), and performance/similarity-based defenses (FLTrust, CC), in addition to the non-robust FedAvg baseline. As shown in Table 1 and Fig. 3, ASAP consistently outperforms existing attacks across different datasets.
>
> ASAP is the only method that can reliably achieve the desired target accuracy. Its key advantages are summarized in Remarks 1, 2, and 3 of the main paper (repeated here for completeness).
>
> **Remark 1: AGR-Agnostic Operation**
>
> Unlike existing AGR-agnostic attacks (LIE, Min-Max, Min-Sum), which still rely on statistical estimation of benign client updates, ASAP is fully independent of both the aggregation rule and benign client information. The ASMC framework treats unknown aggregation effects as system disturbances $\Phi_t$, which are estimated in real time via Fourier series approximation, without requiring any prior knowledge of $F_{\text{AGR}}$ or benign gradient statistics.
>
> **Remark 2: Convergence Speed**
>
> The parameter $k$ acts as a convergence-rate controller, enabling precise manipulation of the tracking error $e_t$. On the sliding surface where $s_t = \dot{s}_t = 0$, solving the differential equation $\dot{e}_t = -k e_t - C$ yields
> $e_t = \frac{1}{k} e_0 e^{-k t} - \frac{C}{k}$.
> This analytical solution shows that $k$ determines the exponential convergence behavior: larger values of $k$ correspond to faster convergence. This property allows ASAP to flexibly modulate convergence speed according to the adversary’s requirements.
>
> **Remark 3: Adjustable Objectives**
>
> The adversary can dynamically adjust the attack objective during ASAP’s execution by appropriately choosing the parameter $C$ in the definition of $e_t$ (as specified in the main paper). When the system reaches equilibrium on the sliding manifold, where both $\dot{s}_t = 0$ and $\dot{e}_t = 0$, the constraint
> $\dot{s}_t = \dot{e}_t + k e_t + C$
> implies the equilibrium relationship $e_t = -C/k$, or equivalently,
> $g_t = \tilde{g} + C/k$.
> This establishes a direct link between $C$ and the final attacked global model, enabling controllable and adjustable attack objectives.
>
>
> **References**
>
> [1] Suiyang Khoo, Juliang Yin, Zhihong Man, and Xinghuo Yu. Finite-time stabilization of stochastic
> nonlinear systems in strict-feedback form. Automatica, 49(5):1403–1410, 2013.
>
> [2] Juliang Yin, Xin Yu, and Suiyang Khoo. Finite-time stability of stochastic nonlinear systems with
> markovian switching. In 2017 36th Chinese Control Conference (CCC), pp. 1919–1924, 2017.
> doi: 10.23919/ChiCC.2017.8027634.
>
> [3] D. Shevitz and B. Paden. Lyapunov stability theory of nonsmooth systems. IEEE Transactions on
> Automatic Control, 39(9):1910–1914, 1994. doi: 10.1109/9.317122.

---

### Official Review · Reviewer_AHWr · 2025-11-01

**Soundness:** 3
**Presentation:** 4
**Contribution:** 3
**Rating:** 6
**Confidence:** 3

**Summary:**

This paper addresses the problem of aggregation-rule agnostic poisoning in federated learning, where existing attacks (e.g., LIE, Min-Max, Min-Sum, FMPA) mentioned in this paper either rely on estimating benign client statistics or require partial knowledge of the server’s aggregation rule, limiting their applicability in realistic settings. To overcome this, the authors propose ASAP, an adaptive sliding-mode control (ASMC)–based attack framework that treats the entire FL process as a nonlinear dynamical system and models unknown aggregation effects as bounded disturbances. This design allows ASAP to precisely steer the global model toward a target accuracy without knowing the aggregation rule or benign updates. Extensive experiments on CIFAR-10, MNIST, and Tiny ImageNet under nine Byzantine-robust defenses show that ASAP achieves 2–5% deviation from target accuracy, while prior AGR-agnostic baselines deviate by 10–40%, and it requires only ~1/40 the communication rounds of Min-Max and LIE. The results demonstrate that ASAP consistently outperforms state-of-the-art agnostic attacks while maintaining full independence from benign information and offering controllable attack speed and precision.

**Strengths:**

++ It introduces a control-theoretic formulation (Adaptive Sliding Mode Control, ASMC) for federated poisoning: a rarely explored direction in FL security research.

++ It provides controllable poisoning: the first to enable precise adjustment of convergence speed and target accuracy under unknown aggregation rules, enabling more stealthy attacks.

++ It derives the attack dynamics from nonlinear system theory, using Lyapunov stability analysis to guarantee finite-time convergence toward an adversarial target. The control law and adaptive estimator are logically consistent and theoretically justified (e.g., sliding manifold design, adaptive law for bounded uncertainty).

++ Ablation studies demonstrate robustness against varying non-IID degrees, client numbers, and sampling rates, consistently outperforming other AGR-agnostic baselines

++ Writings:





Good-structured flow: motivation → theory → algorithm → experiments → analysis



Clear motivation and consistent terminology; key differences from baselines are well-explained

**Weaknesses:**

-- The work doesn’t directly compare to the latest 2024–2025 poisoning attacks (e.g., [1],  [2]), leaving a slight novelty gap in the context of recent research trends.

-- The continuous-time control formulation assumes smooth model dynamics, which may not hold in discrete, stochastic FL updates.

-- (minor) Though broad, experiments are all simulation-based; no real-world FL deployment or asynchronous communication setting is tested.



[1]. PoisonedFL — Model Poisoning Attacks via Multi-Round Consistency (Xie, Fang, Gong; CVPR 2025)

[2]. Data-Agnostic Model Poisoning against Federated Learning: A Graph Autoencoder Approach (Li et al., TIFS 2024)

**Questions:**

Please refer to the Weaknesses.

---

> ### Author Response · Authors · 2025-11-22
> **Response to Reviewer AHWr**
>
> Thank you for your detailed review!
>
> **W1: Comparison with latest 2024-2025 attacks.**
> Due to PoisonedFL (CVPR 2025) and Graph Autoencoder approach (TIFS 2024) not having the controllability to achieve the target accuracy, ASAP is the only method that can achieve the target accuracy. Fundamental reasons are given in Remarks 2 \& 3 of the main paper (re-inserted here for completeness). The proposed control and adaptive laws are shown through Theorem 3.1 (proof in Appendix A) that sliding surface where $ s_t =  \dot{s}_t=0$ is guaranteed to be reached in finite time. Then, both convergence speed and objectives are adjustable:
>
> **Remark 2: Convergence Speed.**
> The parameter $k$ serves as a convergence rate controller, enabling precise manipulation of $e_t$. On the sliding surface where $ s_t =  \dot{s}_t=0$, solving the differential equation $\dot{e}_t = - ke_t - C$, produces $e_t=1/k \cdot e_0^{-kt}-C/k$. The analytical solution reveals that $k$ determines the exponential convergence characteristics: larger values of $k$ correspond to faster exponential convergence rates. This mathematical property enables ASAP to offer flexible convergence speed modulation capabilities.
>
> **Remark 3: Adjustable Objectives.**
> The adversary can dynamically modify attack objectives throughout ASAP execution by appropriately selecting parameter $C$ in $e_t$ as evaluated in Eq. (16). When the system reaches equilibrium on the sliding manifold where both $\dot{s}_t =0$ and $\dot e_t=0$, the constraint $\dot{s}_t=\dot{e}_t + ke_t + C$ results in the equilibrium relationship $e_t  = -C/k $ or $g_t = \tilde g + C/k$.
>
> Compared to the existing experiments under target accuracy 50\% with FedAvg on CIFAR-10 in the table below, the value in the table is the accuracy of different attacks and the difference between the accuracy of the attack and the target accuracy. We will add more detailed comparisons in the updated submission.
> | Target Acc. | PoisonedFL | GAE-based | ASAP |
> |----------------|----------|-------|--------|
> | 50% | 90.01% (80.02%) | 80.26% (60.52%) | 50.87 (**1.74%**) |
>
> **W2: Continuous-time formulation vs. discrete FL.**
> The fundamental reason for using continuous-time analysis is because structured mathematical rules, especially differentiation, chain rules, and so on, are all well established by mathematicians through measure theory. It is therefore valid to analyze the system in continuous time. Similar to the analysis of back-propagation and steepest descent algorithms, when describing back propagation training algorithm, continuous time analysis is used in the book Neural Networks (Haykin 1999) and Learning representations by back-propagating errors (Rumelhart, Hinton, and Williams 1986).
>
> Regarding our Eq. (13), the dynamic model is used to analyze how the malicious clients update over time, not to implement the algorithm. Based on the definition of derivative, the derivative of the malicious model at time $t$, $\dot{g}'_{t,i}$, can be represented as
>
> $$
> \dot{g}'_ {t,i} = \lim _ {\Delta t \to 0} \frac{g ' _ {t+\Delta t,i} - g ' _ {t,i}}{\Delta t}
> $$
>
> Therefore, in practical attack scenarios, the rate of change of the model, $\dot{g}'_{\{t, i\}}$, can be approximated by the difference in values divided by a small time interval, effectively capturing the derivative's behavior in discrete time. Importantly, Eq. (13) is for analysis purposes. We do not need to calculate equation Eq. (13) in actual implementation. By referring to the various fundamental works in machine learning cited above, we believe similarly for our analysis, it’s natural to use continuous time analysis to analyze our proposed algorithm.
>
> **W3: Asynchronous communication.**
>
> To address the asynchronous setting concern, we have added an experiment on CIFAR10 with AlexNet and 50 clients where 10% are malicious and using ASAP. Instead of all attackers being active from round 0, each malicious client $i$ starts attacking at a different epoch $t$ (e.g., the first at epoch 5, later ones staggered by 3 epochs), and behaves benignly before $t$. The server runs the same Trmean aggregation as in the main experiments. We observe that the attack still degrades the model and maintains similar robustness as in the fully synchronized case.
>
> | Target Acc. | 60%|55%| 50% |
> |----------------|----------|-------|--------|
> | Trmean| 62.15% (3.58%) | 52.31% (-4.89%) | 51.74 (3.48%) |
>
>
> **References**
>
> Haykin, S. 1999. Neural Networks: A Comprehensive Foundation. Prentice Hall.
>
> Rumelhart, D. E.; Hinton, G. E.; and Williams, R. J. 1986. Learning representations by back-propagating errors. nature, 323(6088): 533–536.

---

> > ### Comment · Reviewer_AHWr · 2025-11-25
> >
> > Thanks for addressing all my concerns! I will keep my current positive score.

---

### Author Response · Authors · 2025-12-03
**Summary for Area Chair - ICLR 2026 Submission 16711**

**Acknowledgments:**

We sincerely thank the ICLR 2026 Conference Program Chairs for organizing this conference and the Senior Area Chairs for their oversight of the review process. We are particularly grateful to the Area Chair for coordinating our submission review and providing us the opportunity to respond. We deeply appreciate all reviewers (Reviewer AHWr, Reviewer y5Zy, Reviewer yJY3, Reviewer tE3W) for their thorough evaluation, constructive feedback, and insightful questions throughout both review rounds. Their comments have significantly strengthened our work and helped us clarify critical technical aspects. We also thank the ICLR 2026 program committee for their efforts in maintaining the high standards of the conference.

---

**1. Overview**

We propose ASAP, a novel AGR-agnostic model poisoning attack on Federated Learning that achieves **precise control over attack objectives** without requiring knowledge of server aggregation rules (AGRs) or benign client information. Our approach is grounded in **Adaptive Sliding Mode Control (ASMC)** theory and uses Fourier series approximation to estimate unknown system dynamics.

---

**2. Key Contributions**

**2.1 True AGR-Agnosticism**

Unlike existing agnostic attacks (LIE, Min-Max, Min-Sum) that still require statistical estimation of benign client updates, ASAP achieves complete independence from both aggregation rules and benign client information through control-theoretic design.

**2.2 Precise Controllability**

ASAP provides adjustable and flexible control over:
- **Convergence speed** (via parameter $k$)
- **Attack objectives** (via parameter $C$)
- Attackers can dynamically modify targets during training process

**2.3 Theoretical Guarantees**

We prove via Lyapunov function that ASAP guarantees:
- Finite-time convergence to the sliding surface
- Exponential convergence of error to target accuracy
- Robustness against various AGRs

**2.4 Superior Empirical Performance**

Extensive experiments on CIFAR10, MNIST, and Tiny ImageNet demonstrate that ASAP:
- Achieves the closest accuracy to target objectives (lowest $|\varsigma|$ values)
- Requires fewer communication rounds than competing methods
- Consistently bypasses diverse Byzantine-robust defenses

---

**3. Response to Reviewer Common Concerns**

**3.1 Continuous-Time Formulation vs. Discrete FL (All Reviewers)**

**Concern**: FL operates in discrete rounds, yet ASAP uses continuous-time analysis.

**Response**:
- The continuous-time formulation is standard practice in control theory and optimization (similar to backpropagation analysis)
- The derivative $\dot{g} ' _ {t,i}$ is approximated in practice as:  $\dot{g} ' _ {\{t, i\}} = \lim_{\Delta t \to 0} \frac{g' _ {(\{t+\Delta t), i\}} - g' _ {\{t, i\}}}{\Delta t}. $
- The dynamic model is used for analysis, not direct implementation

**3.2 Non-Differentiable AGRs (Reviewer y5Zy, Reviewer yJY3)**

**Concern**: How does Theorem 3.1 hold for non-differentiable AGRs?

**Critical Clarification**:
- **ASAP does NOT differentiate through AGR rules**
- We differentiate the **global model $g_t$** (which is a neural network—always differentiable)
- Lyapunov stability theory applies to non-smooth systems (supported by literature: Shevitz & Paden 1994, and work on stochastic systems)

**3.3 Stealthiness and Detection (Reviewer y5Zy, Reviewer yJY3, Reviewer tE3W)**

**Concern**: Why does ASAP bypass robust AGRs?

**Mechanism Explanation**:
- Traditional attacks produce extreme updates that AGRs detect and discard
- ASAP uses **Adaptive Sliding Mode Control** to generate malicious clients' updates to steer the global model to the desired attack objectives

**Evidence**: Table 1 shows ASAP consistently achieves target accuracy across 9 different AGRs while other attacks fail or are detected.

**3.4 Separate Response to each Reviewer**

In response to the remaining reviewer-specific comments, we have: (i) for **Reviewer AHWr**, added comparisons with recent attacks (PoisonedFL, GAE-based) and an asynchronous FL experiment with staggered malicious participation; (ii) for **Reviewer y5Zy**, detailed the sampling/participation of malicious clients and evaluation of stealthiness; (iii) for **Reviewer yJY3**, clarified the architecture and threat-model mapping; (iv) for **Reviewer tE3W**, justified high target (control), added low target results, clarified Figure 1 as conceptual, reported TP/TN/FP/FN, fixed tables, and explained AGR failure and Fourier convergence.

---

**4. Recommendation for AC**

We have thoroughly addressed all reviewer concerns through clarifications on continuous-time analysis, non-differentiable AGRs, stealthiness mechanisms, and new experiments, corrected tables, typos. The work makes both theoretical and practical contributions that warrant publication at ICLR 2026.

---

### Author Response · Authors · 2025-12-04
**Revised Submission Clarifications and Corrections**

**Summary of Changes**

The revised submission adds a new theoretical subsection in Section 3.3 (lines 377-395) addressing two critical concerns: justification for the continuous-time formulation in discrete FL settings, explaining how derivatives approximate discrete differences, and clarification that ASAP only requires differentiability with respect to model parameters rather than the aggregation rule itself, which is crucial for handling non-differentiable AGRs like Median and Krum.

The experimental section has been significantly expanded in Table 1 with three new extreme target accuracy scenarios (CIFAR10 at 10%, MNIST at 10%, Tiny ImageNet at 0.5% ) to demonstrate ASAP's control capabilities across a wider performance, including near-random-guess baselines. Table 3 in the appendix has been correspondingly expanded with three additional target accuracy levels (CIFAR10 30%, MNIST 50%). Minor text refinements have been made throughout, particularly in the Section 4.2 experimental results discussion.

---

### Meta-Review · Area_Chair_UFiX · 2026-01-06

**Summary:**

This paper proposes ASAP, a control-theoretic, aggregation-rule-agnostic poisoning attack for federated learning, aiming to precisely steer the global model to a target accuracy without knowledge of the server’s aggregation rule or benign updates. Reviewers acknowledged the originality of introducing adaptive sliding mode control into FL security and the strong empirical controllability results. However, the suggested decision is mainly informed by persistent concerns about the soundness of the theoretical justification under non-smooth robust aggregation rules, the lack of a clear mechanistic explanation of why robust defenses fail, and incomplete evaluation of stealthiness and realism.

**Reviewer Concerns:**

In the rebuttal, the authors clarified the continuous-time formulation, added extreme low-target experiments, and expanded empirical results, which strengthen the controllability demonstration and improve presentation. However, several core concerns remain unresolved. Multiple reviewers were not convinced that the Lyapunov-based analysis genuinely applies to non-differentiable robust AGRs such as Median and Krum, noting that the current explanation does not fully reconcile the theoretical assumptions with the actual attack setting. In addition, the paper still lacks a convincing mechanistic account of how ASAP bypasses Byzantine-robust defenses, and the evaluation of stealthiness and detectability remains limited. Questions about the realism of the participation model and the interpretation of the “closest to optimum” objective also persist.

**Reviewer Scores:**

No reviewer indicated a clear positive score change after rebuttal. One reviewer maintained a marginally positive score, while others explicitly kept their below-threshold ratings due to unresolved theoretical and practical concerns. I believe most reviewers would likely maintain their original scores after full discussion.

---

### Decision · Program_Chairs · 2026-01-26

Reject